

Observations and hypotheses related to low to middle free tropospheric aerosol, water vapor and
altocumulus cloud layers within convective weather regimes: A SEAC[4]RS case study
Jeffrey S. Reid[1]*, Derek J. Posselt[2], Kathleen Kaku[3], Robert A. Holz[4], Gao Chen[5], Edwin W.
Eloranta[4], Ralph E. Kuehn[4], Sarah Woods[6], Jianglong  Zhang[7], Bruce Anderson[5], T. Paul Bui[8],
Glenn S. Diskin[5], Patrick Minnis[5], Michael J. Newchurch[9], Simone Tanelli[2], Charles R. Trepte[5],
K. Lee Thornhill[5], Luke D. Ziemba[5]
[1]US Naval Research Laboratory, Marine Meteorology Division Monterey CA
[2]Jet Propulsion Laboratory, Pasadena CA
[3]General Dynamics, Naval Research Laboratory, Monterey CA
[4]Space Sciences Engineering Center, University of Wisconsin, Madison WI
[5]NASA Langley Research Center, Science Directorate, Hampton VA
[6]SPEC Inc. Boulder CO.
[7]University of North Dakota, Grand Forks, ND
[8]NASA Ames Research Center, Mountain View, CA
[9]Atmospheric Science Department, University of Alabama in Huntsville, Huntsville, AL
Key points:
1)  Highly sensitive lidar and aircraft observations reveal thin aerosol detrainment layers
19          from convection and their associated altocumulus clouds.
2)  At $0^o$C there is a proclivity for aerosol and water vapor detrainment from storms, in
21          association with melting level Altocumulus shelves.
3)  Detraining particles undergo chemical and microphysical transformations with enhanced
23          nucleation in cleaner environments.

*Corresponding author: Jeffrey S. Reid (jeffrey.reid@nrlmry.navy.mil)





**Abstract:** The NASA *Studies of Emissions & Atmospheric Composition, Clouds & Climate*
*Coupling by Regional Surveys* (SEAC⁴RS) project included goals related to aerosol particle
lifecycle in convective regimes. Using the University of Wisconsin High Spectral Resolution
Lidar system at Huntsville, Alabama USA and the NASA DC-8 research aircraft, we investigate
the altitude dependence of aerosol, water vapor and Altocumulus (Ac) properties in the free
troposphere from a canonical August 12, 2013 convective storm case as a segue to a presentation
of a mission wide analysis. It stands to reason that any moisture detrainment from convection
must have an associated aerosol layer. Modes of covariability between aerosol, water vapor and
Ac are examined relative to the boundary layer entrainment zone, $0^{\circ}C$ level, and anvil, a region
known to contain Ac clouds and a complex aerosol layering structure (Reid et al., 2017).
Multiple aerosol layers in regions warmer than $0^{\circ}C$ were observed within the PBL entrainment
zone. At $0^{\circ}C$ there is a proclivity for aerosol and water vapor detrainment from storms, in
association with melting level Ac shelves. Finally, at temperatures colder than $0^{\circ}C$, weak aerosol
layers were identified above Cumulus congestus tops ($\sim 0^{\circ}C$ and $\sim -20^{\circ}C$). Stronger aerosol
signals return in association with anvil outflow. In situ data suggest that detraining particles
undergo aqueous phase or heterogeneous chemical or microphysical transformations, while at the
same time larger particles are being scavenged at higher altitudes leading to enhanced nucleation.
We conclude by discussing hypotheses regarding links to aerosol emissions and potential indirect
effects on Ac clouds.
**Plain language summary:** In studies of the vertical transport of air pollution by clouds as well
as pollution's subsequent impact on those clouds the scientific community often focuses on
clouds with bases at the planetary boundary layer (such as typical fair weather cumulus) and the
outflow from thunderstorms at their tops. However, new highly sensitive lidar systems
demonstrate complex aerosol features in the middle free troposphere. Aerosol layers formed in
convective outflow are explored and are shown to have strong relationships to mid-level
tropospheric clouds, an important but difficult-to-model or monitor cloud regime for climate
studies.





## 1.0    Introduction

Much of the focus of aerosol-cloud radiation studies (i.e., the first indirect effect) has been on either Planetary Boundary Layer (PBL) Stratocumulus (Sc) or Cumulus clouds (Cu, e.g., Twomey et al., 1977 and many subsequent citations), or the injection of aerosol particles and their precursors into the upper troposphere/lower stratosphere by deep precipitating convection (Cb, e.g., Pueschel et al., 1997; Kulmala et al., 2004; Waddicor et al., 2012; Saleeby et al., 2016), pyro-convection (e.g., Fromm et al., 2008; 2010; Lindsay and Fromm 2008) and volcanic activity (e.g., Jensen and Toon 1992;  DeMott et al., 1997; Amman et al., 2003). However, there is a third important but often overlooked aerosol-cloud system related to mid-level clouds. Altocumulus (Ac) clouds in the lower to middle free troposphere (LMFT) are generated by numerous mechanisms (e.g., synoptic forcing, gravity waves, orographic waves), but are particularly prevalent in convective regimes (Heymsfield et al., 1993; Parungo et al., 1994; Sassen and Wang, 2012). Indeed, the above authors and others (e.g., Gedzelman, 1988) note these cloud types receive comparatively little attention in the scientific community relative to their importance. Forecasters sometimes ignominiously note the presence of Ac in convective environments as "midlevel convective debris." Yet, Cloud-Aerosol-Lidar with Orthogonal Polarization (CALIOP) and CloudSat retrievals attribute to Ac as much as 30% area coverage in Southeast Asia and the summertime eastern continental United States (e.g., Zhang et al., 2010; Sassen and Wang, 2012; Zhang et al., 2014). This is in agreement with observer-based cloud climatologies (e.g., Warren et al., 1986; Warren et al., 1986).

A long-standing hypothesis by Parungo et al., (1994) suggested that globally increasing aerosol emissions would lead to higher mid troposphere aerosol loadings, in turn enhancing Ac reflectance and perhaps Ac lifetime. This is plausible, as Kaufman and Fraser (1997) mistook Sc and Ac clouds for Cumulus mediocris (Cu) in their analysis of cloud reflectivity and lifetime impacts by biomass burning particles (Reid et al., 1999). Lidar studies by Schmidt et al., (2015) showed significant sensitivity of cloud droplet size distributions to aerosol particles near cloud base. Yet Ac's diurnal cycle, covariance with other cloud types including cirrus during convective detrainment, and sometimes tenuous cloud optical depth make Ac clouds difficult to characterize and monitor. In an inter-comparison study for Southeast Asia, Reid et al., (2013) found more diversity in midlevel cloud fractions between satellite products than at any other





level. Likewise, large scale models tend to underestimate Ac formation and liquid water content
(Barrett et al., 2017).
Ac cloud are prevalent in many forms such as: castellanus, an indicator of midlevel instability;
mountain wave lenticularis; and translucidus (or colloquially mackerel sky). One class of Ac
clouds is colloquially referred to as shelf clouds, caused in part by detrainment at mid-level from
deep convection (Fig. 1(a); see Johnson et al., 1999; Yasunga et al., 2006). These are not
assigned their own genus in the International Cloud Atlas (Cohen et al., 2017), but the generic Ac
is recognized as associated with the spreading of convective elements at a stable layer.
Ac shelves often form at $0^{o}$C from deep convection or in association with mid-level inversions
(e.g., Johnson et al., 1996, 1999; Riihimaki et al., 2012). A primary production mechanism is
thought to be related to the formation of $0^{o}$C stable layers initiated by the melting of falling
frozen hydrometeors and enhanced condensation to compensate for the cooling (Posselt et al.,
2008; Yasunga et al., 2008). Hydrometeor evaporation processes discussed in Posselt et al (2008)
have likewise been hypothesized to help form the inversion. This results in a thin cloud feature
forming just below the inversion. Shelf-like Ac from towering cumulus (TCu) are also frequently
observed (Fig. 1(b)), and may be related to the detrainment of overshooting tops around regional
$0^{o}$C stable layers formed by surrounding convection (Johnson et al., 1996), or upper level
subsidence. Combined Ac and associated Alto stratus (As) coverage can be high in convectively
active regions (Fig. 1(c)). Ac can also form overnight from the residual PBL in convective
environments and then burn off during the day (Fig. 1(d); Reid et al., 2017) or during fair
weather conditions just ahead of more active weather (Figure 1(e)). Ac by mesoscale lifting is
also common. Although sometimes geometrically thin with low liquid water contents, Ac can
generate copious virga (Figure 1(f)).
Compared to other cloud species, the relationship between LMFT aerosol layers and Ac clouds
has a small literature base. The largest fraction of papers relate to lidar observations of smoke
and dust as ice nuclei (IN) in mixed-phased alto clouds (e.g., Hogan et al., 2003; Sassen et al.,
2003; Wang et al., 2004; Sassen and Khvorostyanov, 2008; Ansmann et al., 2009; Wang et al.,
2015). However, cloud condensation nuclei (CCN) budgets for these cloud types have not been
studied in detail with in situ observations, particularly for entirely liquid clouds. The complex
mixed-phase nature of alto-level clouds and stratiform precipitation coupled with their thin



nature and low updraft velocities (Schmidt et al., 2014) likely lead to sensitivity to even small
perturbation in CCN concentration (Reid et al., 1999; Schmidt et al., 2015; Wang et al., 2015).
Clouds can serve as aqueous phase reactors of gas and aerosol particle species, even hosting
nucleation events (Hegg et al., 1991), while evaporating droplets and precipitation leave residual
aerosol particles. Given that Ac clouds are observed to have a strong impact in shortwave solar
radiation (Sassen and Khvorostyanov, 2007), the hypotheses of Parungo et al., (1994) are worthy
of consideration despite initial skepticism (e.g., Norris 1999). Only now are the tools becoming
available to quantitatively investigate further.
Observing the aerosol-Ac environment is challenging. The scarcity of data for alto-level aerosol
layers in the convective regimes where Ac clouds often form, combined with the contextual or
sampling biases inherent for the in situ observations of such layers and sun-synchronous polar-
orbiting aerosol observations, obscure the true importance of LMTF aerosol layers in
atmospheric aerosol lifecycle and Ac cloud physics. An opportunity for study arose with the
summer 2013 NASA *Studies of Emissions & Atmospheric Composition, Clouds & Climate*
*Coupling by Regional Surveys* (SEAC[4]RS; *Toon et al.,* 2016) field mission. For SEAC[4]RS, the
NASA DC-8, NASA ER-2 and Spec Inc Lear-25 aircraft were deployed along with ground assets
including the University of Wisconsin Space Science and Engineering Center (SSEC) High
Spectral Resolution Lidar (UW-HSRL) to examined the aerosol and cloud environment of the
summertime eastern United States (Toon et al., 2017; Reid et al., 2017). These observations
allowed for comprehensive measurements of the structure and microphysical properties of local
convectively generated LMFT aerosol layers.
SEAC[4]RS provided a valuable but complex dataset-especially in the vicinity of active
convection. To simplify the analysis, this paper provides a case study of the covariability
between aerosol layers and LMFT Ac clouds in convective environments using observations
collected on August 12, 2013. This day was chosen due to the isolated regional nature of the
convection that occurred, and availability of ground based lidar and airborne DC-8 sampling.
This analysis will provide context for further exploration of the SEAC[4]RS datasets.
For this analysis we define Ac consistent with the WMO definition (Houze 1993; WMO
https://cloudatlas.wmo.int/clouds-definitions.html last accessed Mar 2018) of mid-altitude (2-7
km) clouds that are a) liquid or mixed phase, and b) decoupled from direct surface forcing. We





begin with a brief description of data sets used in the remainder of the paper (Section 2). We then
provide an overall narrative of the meteorological situation on August 12 (Section 3) followed by
an analysis of UW-HSRL (Section 4) and data collected from a nearby storm by the DC-8
(Section 5). In the paper's discussion (Section 6), we explore commonalities in the two datasets,
and further explore hypotheses of LMFT layer characteristics, their origins, and relationships to
Ac clouds to set the stage for subsequent papers. A final summary and conclusions are presented
in Section 7.
## 2.0    **Data and Methods**
The analysis presented here centers around the August $12^{th}$ 2013 SEAC$^4$RS airborne research
flight based out of Ellington Field, Houston TX (Toon et al., 2016). The Ellington deployment
for the SEAC$^4$RS mission was conducted from August $12^{th}$ -September $23^{rd}$ with three research
aircraft (NASA DC-8, NASA ER2, SPEC Learjet 25), an extensive ground network including
AERONET sun photometers (Holben et al., 1998; Toon et al., 2016), and the deployment of the
UW-HSRL to Huntsville (Reid et al., 2017). Comprehensive descriptions of the field assets is
provided in this section's cited papers; here we provide a short summary of datasets used in this
analysis.
### 2.1    UW-HSRL Deployment to Huntsville
LMFT aerosol and cloud layers were monitored by a 532 nm UW-HSRL system, deployed by
the NASA Cloud-Aerosol Lidar and Infrared Pathfinder Satellite Observations (CALIPSO)
science team to enhance monitoring at the Regional Atmospheric Profiling Center for Discovery
(RAPCD) lidar facility at the UAH National Space Sciences Technology Center (NSSTC)
building (-34.725 $^o$ N; 86.645$^o$ W), from June 18 to November 4, 2013. The RAPCD facility is
located on the western side of the city of Huntsville at an elevation of ~220 m. Including
building height, the lidar transmitter was situated at 230 m above mean sea level (MSL). Overall
the local terrain is flat, with the exception of a line of hills protruding an additional 200-350 m
and located 10-15 km to the east and southeast. The UW-HSRL was hardened for continuous
use, and collected contiguous aerosol backscatter and depolarization data every 1 minute at 30 m
vertical resolution. The only significant notable outages were from August $20^{th}$ -$22^{nd}$ and
September $13^{th}$-$17^{th}$. UW HSRL observations can be visualized and downloaded through the
SSEC HSRL web page (http://hsrl.ssec.wisc.edu/), last verified in February 2019.



The UW-HSRL was able to extract the aerosol backscatter profile to very high fidelity. Unlike
more common elastic backscatter lidar measurements that must de-convolve a combined
molecular and aerosol signal in an inversion, HSRL systems can separate a line broadened
molecular backscatter signal from the total backscatter signal via a notch filter (Eloranta et al.,
2005, 2014; Hair et al;, 2008). The difference is used to calculate aerosol backscatter. For this
deployment the UW HSRL performed with a precision in aerosol backscatter of better than $10^{-7}$
$(m\ sr)^{-1}$ for a 1 minute average, and $10^{-8}\ (m\ sr)^{-1}$ for 15 minute averages. In comparison, Rayleigh
backscattering is $1\times10^{-6}\ (m\ sr)^{-1}$ at 4 km, and $5\times10^{-7}\ (m\ sr)^{-1}$ at 10 km. Thus at 15 min averaging,
precision is likewise better than 1 to 5% of Rayleigh.
By calculating the slope of the returned molecular scattering, aerosol light extinction can be
directly calculated. However, as described in Reid et al., (2017), there are several caveats. First,
there must be significant enough signal to calculate the slope; in this instrument, extinction must
be greater than 0.1 $km^{-1}$. Second, one must account for an "overlap correction" in the near field,
accounting for the fact that the telescope is not fully in focus until a range of about 4.5 km from
the system. The signal below the 4.5 km level appeared to vary in time, sometimes hourly, during
the daytime. Consequently, for the altitude range we will study here, it is best to rely on aerosol
backscatter. Noting that extinction is simply the aerosol backscatter times the lidar ratio ($S_a$),
here we assume a lidar ratio of 55 $sr^{-1}$ as a baseline (Reid et al., 2017). Expected deviations from
this baseline are discussed in the Results and Discussion sections.
In addition to the lidar, several other deployments to the UAH site are used here. Most notably,
UAH was a Southeast American Consortium for Intensive Ozonesonde Network Study
(SEACIONS) release site (http://croc.gsfc.nasa.gov/seacions/, last accessed December 17, 2018).
Forty sondes were released between August $6^{th}$ and September $21^{th}$, 2013, at 18:00-19:00
Z/13:00-14:00 CDT to coincide with early afternoon boundary layer conditions, mid-flight
airborne activity, and the NASA A-train overpass. For August 12, 2013, the release time was
13:42 CDT, and is used here for situational awareness and the mapping of cloud and aerosol
layers to their temperatures.
2.2    The SEAC$^4$RS DC-8 Operations
The DC-8 conducted 24 flights with patterns that covered the Western United States through the
Southeastern United States (SEUS) and into the Gulf of Mexico. Flight patterns often included





three primary relevant components. 1) A ~100 km curtain wall pattern with multiple flat flight
levels from 5 km to the near surface to collect free troposphere, entrainment zone, cloud base and
near surface samples; 2) saw toothed transits to monitor the lower troposphere for chemistry
applications; and 3) spirals in the vicinity of developing deep convection. Flight restrictions in
the vicinity of Huntsville prevented vertical profiles directly over the UW-HSRL. Nevertheless,
the DC-8 had ample opportunity to sample the SEUS' LMFT environment, in particular for the
case of August 12, 2013 examined here.
The DC-8 hosted its most comprehensive instrument suite ever to support the chemistry,
convection, radiation, and upper troposphere/lower stratosphere (UTLS) science goals and
customers. However, for the particular test case and application examined here, there are several
caveats worth noting. While the ground-based UW HSRL can detect the fine aerosol structure in
convective environments and in the vicinity of Ac clouds, generating in situ observations to
correspond to this structure is difficult. At flight speeds of ~120-150 m s$^{-1}$, the DC-8 is only in a
detrainment patch for a few seconds, causing difficulty in differentiating small-scale aerosol
features. Further, the massive payload of the DC-8, although comprehensive, also leads to
functional problems as instrument calibration, maintenance, or scanning cycles were not
synchronized. Shattering effects of liquid cloud droplets and ice further disrupted the sampling of
the very near cloud environment. Thus, one cannot retrieve full complement of all data for an
entire profile or flight component, let alone for individual features that the DC-8 might observe
for less than 10 seconds. While the DC-8 carried a lidar system of its own, stand-off distances
from the aperture and cloud heterogeneity prevented its use in this particular analysis.
Nevertheless, the DC-8 hosted a number of instruments that can provide a valuable view of the
overall aerosol and cloud structure in the August 12$^{th}$ 2013 convective environment which can be
coupled with the lidar observations. These key instruments are listed here.
1)   State variables: Navigation was derived from DC-8 housekeeping variables. Pressure,
temperature and winds were measured by the NASA Ames Meteorological Measurement System
(MMS, Scott et al., 1990). Moisture related variables were derived from the NASA Langley
Diode Laser Hygrometer (DLH, Podolske et al., 2003; Livingston et al., 2008).



2)   Aerosol physical and optical properties: Baseline aerosol number, size, and optical
properties were derived from the Langley Aerosol Research Group Experiment
(https://airbornescience.nasa.gov/instrument/LARGE, Ziemba et al., 2013; Corr et al., 2016)
instrument set, which included continuously sampling nephelometer, CN, and optical particle
encounters. The LARGE package monitored aerosol particles from ultrafine CN to an inlet cut
point of ~3.5 μm, and units reflect volumetric scaling to a standard temperature and pressure of
$20^{o}C$ and 1013 hPa. To prevent any possible cloud water or precipitation shattering effects on the
aerosol instruments, CN, nephelometer, and LAS data was heavily cloud screened with data
points removed for one second before the arrival and two seconds after the exit of any cloud with
LWC>0.005 g $m^{-3}$.
3)   Aerosol chemistry: Aerosol chemistry was evaluated using data from the CU aircraft HR-
AMS (Canagaratna et al., 2007; Dunlea et al., 2009; http://cires1.colorado.edu/jimenez-
group/wiki/index.php/FAQ_for_AMS_Data_Users last accessed Mar 2018) that reports the
composition of submicron non-refractory particles. Reported O/C and OA/OC ratios from this
instrument were derived using the updated calibration of Canagaratna et al (2015). Unlike single
particle instruments, the AMS is fairly insensitive to inlet artifacts during cloud penetration. Data
points that were flagged as being potentially impacted by such artifacts (by monitoring excess
water and/or zinc in the aerosol mass spectrum) were removed prior to analysis.
4)   Cloud properties: Cloud detection properties were derived from the SPEC microphysics
package (e.g., Lawson 2011; Lawson et al., 2001; 2006; 2010), in particular the Fast Cloud
Droplet Probe (FCDP) which provided the core cloud liquid water product and the 2D-2 for ice
identification.
5)   Gas chemistry: While the DC-8 carried comprehensive gas chemistry instrumentation, for
this overview case study we rely on CO from the Differential Absorption CO measurement
(DACOM, Sachse et al., 1987; McMillan et al., 2011).
2.3   Ancillary datasets:
In the analysis presented here multiple data sets were examined, but for brevity are not shown in
detail here. Regional meteorology was diagnosed through a combination of NEXRAD radar
(NOAA NWS, 1991), GOES-13 geostationary and MODIS satellite datasets and models.
Baseline meteorology was provided by a Coupled Ocean Atmosphere Mesoscale Prediction



System (COAMPS®) analysis including NEXRAD precipitation and wind assimilation (Zhao et
al., 2008; Lu et al., 2011). Operational MODIS aerosol (MOD/MYD04, Levy et al., 2013) and
cloud (MOD/MYD 06, Platnick et al., 2003; 2016) were also used. Geostationary imagery was
generated at Space Sciences and engineering center with cloud products generated by Minnis et
al. (2008). Regional aerosol concentrations were taken from South Eastern Aerosol Research and
Characterization (SEARCH, Edgerton et al., 2015) and Chemical Speciation Network (CSN),
and Aerosol Robotic Network (AERONET, Holben et al., 1998) sun photometer data. Back
trajectories were utilized from HYSPLIT (Stein et al., 2015).
**3.0 Regional context for the August 12[th] case**
Analysis of the August 12, 2013 case study is greatly aided by context provided by a regional
weather analysis guided by satellite and lidar observations. A more detailed meteorological
analysis is provided in Supplemental Appendix A. In short, on August 12, 2103 the SEUS was in
a fair weather summertime convective regime, with copious small convection, congestus and
isolated CBs. Images of the cloud field from MODIS and on-aircraft photography are provided
in Fig. 2 (including MOD/MYD cloud top temperatures), along with the afternoon radiosonde
sounding at UAH in Fig. 3 (release 18:40 GMT; 13:40 local CDT time), including (a)
temperature and dewpoint; (b) water vapor mixing ratio; and (c) wind speed and direction. The
diurnal pattern of convection is also provided in NEXRAD composite radar images taken
throughout the day, which are provided in Appendix Fig. A.2.
By daybreak on August 12[th], the convection of the previous day had largely subsided over
Alabama (Figs. A.2(b) and (c)). Northern Alabama experienced developing Cu and Ac, with
cirrus (Ci) intermixed to the north in the morning hours (e.g., Terra MODIS 16:00 UTC, Figs.
2(a) and (c)) in association with the stationary front. A large area of optically thin ~0$^{o}$C clouds,
presumably melting level Ac, extended southward from the more convectively active regions to
the northwest. Cloud fractions outside of the cirrus domain ranged from 70-90%. Just before the
Terra overpass, isolated convection was initiated throughout the region, including several cells
north and east of the UAH site. By early afternoon (Aqua MODIS 19:14 UTC, Figs. 2(b) and A.2
(d) and (e)), isolated precipitating cells were widespread across the region. At the same time,
cloud fractions diminished significantly, with a notable reduction in mid-level Ac (yellow
colors). Low level cloud fractions diminished up to ~60%, but there were larger numbers of
isolated and higher-topped TCu.



Using the DC-8 forward-looking cameras during its flight on August 12, ~21:16 UTC, allows us
to categorize the cloud types and heights of the cloud bases and cloud tops of the observed
clouds at the time of the flight (Fig 2e-h). Forward camera images of the environment very near
the deepest convection are provided in Fig. 2(f), and (g), respectively, with a final nadir image of
the Ac field departing the Cb in Fig 2(h). TCu and Cbs were more isolated, relative to the Ac,
forming in association with the remnant outflow boundaries from previous storms, rather than in
organized and sustained lines. Clearly visible in Fig 2(e) is a cloud base delineating the mixed
layer and the PBL entrainment zone at ~1.5 km, corresponding well to the UAH sounding. This
entrainment zone was populated by Cumulus humilis (CuHu) to Cu with tops based on the DC-8
DIAL HSRL in the 1.5-3.8 km AGL range, functionally defining the top of the PBL. Larger Cu
occasionally rose to as high as 4-4.5 km, or to roughly the $0^{o}$C level from the sounding. TCu rose
to 6-6.5 km, with isolated Cb tops at 12 km. Between the PBL top and the Cb anvils, layers of Ac
clouds were prevalent. Some of these Ac clouds are related to mid-level detrainment from Cbs,
others are clearly emanating near the tops of TCu (e.g., Fig 2(f)-(h)). Near surface haze was also
visible, with Aqua MODIS and AERONET  reporting 550 nm AOD on the order of 0.25-0.35.
Reported $PM_{2.5}$ was on the order of ~8 μg m$^{-3}$.
At the time of the early afternoon UAH radiosonde release, the sounding was typical for the area
for a moderately unstable convective meteorological regime (Fig. 3), with the mixed layer and
top inversion at 1500 m MSL (1280 AGL; Fig. 3(a)). Water vapor mixing ratio (Fig. 3(b)) was
constant, as expected in the mixed layer, falling off rapidly with altitude above, and with small
perturbations associated with temperature inversions. Winds were near constant at $250^{o}$ above
the mixed layer, and with steady increases to 12 m s$^{-1}$ at the $0^{o}$C melting level at 4.6 km
providing only a modest amount of shear (Fig 3(c)). Derived CAPE from the UAH sounding was
1650 J kg$^{-1}$ (moderate instability) consistent with TCu to isolated Cb development. As discussed
n the next section, the corresponding HSRL aerosol backscatter profiles for this release are in Fig
3(d)).
4.0 **Results I: HSRL observations**
While the above analysis qualitatively describes the nature of the cloud fields, the time series of
aerosol backscatter and depolarization from the UW-HSRL from August 12$^{th}$, 0:00 UTC through
Aug 13, 09:00 UTC (Fig. 4 (a) and (b), respectively) provides a quantitative representation of the



intricate regional aerosol and cloud environment. Lidar data in Fig. 4 was averaged over 1
minute intervals and over 30 m vertical layers, and represents a time period that extended from
local sunset of August 11th through daybreak on August 13th. Included for reference are
ceilometer-like cloud bases identified in the lidar data for liquid and ice clouds (Fig 4(c)), with
associated geostationary derived cloud tops. Recall, key temperature, water vapor and wind
levels included from the August 12th, 18:40 UTC SEACIONS radiosonde release are further
provided in Fig. 3(a), (b) and (c) respectively and HSRL aerosol backscatter profiles within +/-
3-hours in (d). Temperature levels from this release are included in Fig. 4.  Likewise, mean and
individual aerosol backscatter profiles (every other 5 minutes average, 30 m resolution) are
included in Figure 3(d) for the two hours after the sounding when the DC-8 was sampling
northern Alabama.
The meteorology and aerosol profiles depicted in Fig. 4 show considerable fine scale structure in
cloud and aerosol features. Considered in concert with Fig. 3, Fig. 4 indicates this day is
consistent with the description of the convective environment in Reid et al., (2017) for a similar
August 8th 2013 case. Thus the description of the overall nature of the aerosol environment does
not need to be repeated here in detail, other than to identify the key layers. During the two hour
period surrounding the 18:40 UTC radiosonde release, there is:  1) A mixed layer that extends
from the surface to 1500 m AGL, identifiable by constant $\omega_v$ (Fig. 3b) and an increase in aerosol
backscatter in height due to increases in RH with height and hence hygroscopic growth (Fig. 3(d)
and 4(a));  2) Above the mixed layer inversion lies the entrainment zone, including visible
detrainment layers;  3) As discussed above and shown in Fig. 2(e), the top of the PBL is
ambiguous as it relates to cloud tops in a heterogeneous cloud field, but a clear reduction in
aerosol backscatter is visible at 4 km, likely related to the tops of regional Cu;  4) A second drop
in aerosol backscatter occurs at the $0^oC$ melting level with on this day, 5) a final aerosol layer
between 6-7 km which, as we discuss later, may be associated with cloud top detrainment from
TCu. Assuming a baseline $S_a$= 55 sr$^{-1}$ as derived by Reid et al., (2017) an aerosol backscatter of
$1\times10^{-6}$ (m sr)$^{-1}$ (yellow) is equivalent to an aerosol extinction of 0.055 km$^{-1}$. Integration of
aerosol backscatter from the surface to 10 km for cloud free periods with this lidar ratio suggests
a 532 nm AOD of ~0.17, dropping to 0.12 later in the day, identical to AERONET.



Moving from the sonde release to the whole period shown in Fig 4, the above description of the
thermodynamic and aerosol state of the atmosphere holds for the day. Clouds and precipitation
are clearly visible in the aerosol backscatter color scales as dark red (backscatter $>10^{-4}$ (m sr)$^{-1}$.
Comparing aerosol backscatter with depolarization  for the whole column (Fig. 4(a) and 4(b)),
clouds dominated by ice are easily identifiable from liquid by depolarization values above 40%
(Sassen, 1991), although as discussed later in association with DC-8 observations, low
depolarization does not exclude the presence of ice. Large liquid water drops can also depolarize
the lidar signal and signify heavy precipitation, and are thus annotated on Fig. 4(a). Yellow
highlight boxes of interesting cloud and aerosol phenomenon are marked on Fig. 4(a), with
corresponding enhancements of key features in Fig. 5 derived from 10 second, 7.5 m data.
Finally certain cloud types are annotated including Ac, Sc, and Ci.
Expanding the analysis to include the early evening of the previous day, radar and satellite data
(Fig A.1 and A.2) indicated multiple Cbs at various states of lifecycle were within 15-30 km of
the UAH lidar site. Consequently, cirrus (notable by their high depolarization) was detected
through Aug 12, 2013 7:00 UTC (2:00 CDT) with "bases" for virga  or ice falls between 8 to 13
km, or -35 to -57$^o$C. Given that homogenous ice nucleation can begin at -37$^o$C, except in the
most extreme conditions, at these temperatures water tends to be ice (Pruppacher and Klett,
1997; Campbell et al., 2015). Virga is observed at cloud bases at ~4.5 and ~8.5 km MSL,
highlighted in Fig. 5(a). Using depolarization, we can see the upper cloud at 8.5 km and -25$^o$C
has ice virga emanating from super-cooled liquid water in classic Ac fashion. The cloud base at
4.5 km and 0$^o$C is entirely liquid by lidar observation, although we expect mixed phase processes
at work above where the lidar beam was attenuated. This behavior in combination with local
NEXRAD radar data suggests this lower cloud feature is stratiform precipitation from the anvil
of a decaying system.
In the morning of August 12 until just after daybreak (sunrise ~13:05Z; 6:05 CDT), a strong
aerosol return was visible centered on the 1-1.5 km MSL/0.8-1.3 km AGL range, likely a residual
layer from the previous days PBL mixed layer (ML, to 1.2 km), or entrainment zone (EZ, ~2.5
km). This residual layer may have been transported from the east, but also may be a result of
nighttime cooling and enhanced relative humidity and particle hygroscopicity. Morning
Stratocumulus are embedded in this layer and small liquid water Ac cloud returns are also visible



in the morning (inset box Fig. 5 (b)), at 5:00 UTC at ~6 km (-7$^{o}$C), 10:00 UTC 4 km (5$^{o}$C), with
the strongest returns at the 4.7 km 0$^{o}$C melting level at 12:00 UTC. These clouds likely originate
from convective detrainment of water vapor, such as from melting level detrainment of
convection (e.g., Fig. 1(a) & (b)) or from the tops of TCu clouds, sustained by cloud cooling.
Associated with these clouds are clearly visible individual pockets of aerosol particles on the
order of a few hundred meters high and 15-30 minutes in duration. With backscatter returns on
the order of 1 to 5x10$^{-7}$ (m sr)$^{-1}$, such features are <5% of Rayleigh backscatter and demonstrate
the Ac are embedded in larger aerosol features. At wind speeds of 5-10ms$^{-1}$, these pockets are
between ~5-20 km wide.
In the early morning hours local time, tenuous clouds are also observed at 1 km within the ML
residual layer, likely nighttime radiatively driven Sc. By local daybreak, CuHu begin to more
systematically form at ~1 km due to solar heating at the surface, with cloud base heights
increasing to 1.5 km as the ML and PBL develop throughout the morning to early afternoon LST
(inset Fig. 5 (c)). Clouds also formed at daybreak at 1.5 km inside a PBL residual aerosol layer.
At this height, above the CuHu, these clouds are decoupled from surface forcing and are
optically thin suggesting they are Ac, even though they share their initial formation physics with
Sc earlier in the day. More interestingly, a second Ac deck formed shortly thereafter, with 2-2.5
km MSL bases that increased in height with time through the morning to a maximum height of
3.7 km (5.5$^{o}$C), collinear with the depth of the mixed layer. These are highlighted in inset box
Fig. 5(c). Based on geostationary imagery, and as demonstrated in the comparison of Figure 2(a)
to (b), these clouds evaporated at noon local time, presumably under solar radiation. This
situation is similar to the case of Fig. 1(f). Interestingly, aerosol layers between the PBL clouds
and the Ac are also visible forming late morning at ~15:30 UTC, and increasing with height with
the developing PBL and the Ac clouds above. Cirrus also begins to advect over the site by
afternoon, largely detraining from thunderstorms to the north and west (Fig. A2 (b)).
By 23:00 UTC, a mature phase Cb spawned by the outflow of the storm sampled by the NASA
DC-8 4-5 hours earlier arrived at Huntsville, bringing showers to moderately heavy rain. The
remnants of the storm extend through the next day, producing Ac visible from August 13, 0:00
to 3:00 UTC between the 4.5 km melting level and 7 km (-12$^{o}$C) and (Fig. 5(d)). These clouds,
most likely local in origin, are often categorized as convective debris Ac by the forecasting and





aviation community-an indicator of multi-level detrainment in the convective environment. An
aerosol layer exists to approximately the 4.5 km 0°C melting level capped by Ac. Additional Ac
exist above these embedded in faint but clearly visible aerosol layer features. Unlike the aerosol
pockets earlier in the day, these features are much more limited in extent, no more than 200-300
m in depth.
As the PBL collapses during the evening, it leaves a 1 km AGL residual layer not unlike those
present a day earlier. A final set of light showers from a decaying system occurs in after the early
morning of August 13 at 7:30 UTC (Figure 5(e)). With another clear melting level visible in the
depolarization data, this is likely residual stratiform precipitation like at the beginning of the
timeseries. Similar to the beginning of the time series, ice precipitation from super-cooled liquid
water clouds was also present.

**5.0 Results II:  DC-8 Observations of an August 12, 2013 storm outflow**

The HSRL gives an excellent depiction of the overall aerosol backscatter and cloud phase over
the course of the day, but it lacks the ability to provide microphysical and chemistry information
on the aerosol particles themselves. For this purpose, we utilize measurements on the DC-8 that
flew in the region on this day. The flight pattern on August 12[th] included a curtain wall over the
Gulf of Mexico, saw tooth transit to a curtain wall over northeastern Alabama, and more saw
tooth patterns to a spiral on the downwind side of deep convection developing over the
northwestern corner of Alabama. This last maneuver in northern Alabama is marked on Fig. 2(b),
and provided the day's only complete tropospheric profile. Being on the downwind side of the
storm's trajectory, this profile also gives a snapshot of the aerosol environment detraining from
an isolated storm being fed by a polluted boundary layer. As the storms later passed over
Huntsville, observations collected by the DC-8 also provided context for the UW HSRL lidar
observations described in Section 4. Fig. 2 includes forward and nadir images of the overall
environment. However, the most representative depiction of the midday to early afternoon
environment is provided in Fig. 2(e), taken at 10 km altitude just as the DC-8 started its return
from sampling the storm. The region had a deck of CuHu and Cu with bases at 1.4 km MSL/~1.2
km AGL, delineating the PBL's mixed layer from its entrainment zone. As mentioned, the PBL
top was more ambiguous, and is functionally defined by the tops of these clouds at ~2.5-4 km
(e.g., Fig. 2(a)). TCu were observed, overshooting above the 0°C level, as were scattered Cbs





with tops at ~12-13 km. Ac were prevalent on the horizon, detraining both from overshooting
TCu and midlevel of Cbs.
Profile variables collected by probes on the DC-8 during the spiral initiated at 19:10:30 are
provided in Figure 6. Included are (a) temperature and dewpoint (of liquid water) and tracer
species (b) water vapor mixing ratio ($\omega_v$) and CO. To depict particle scattering (c) provides the
DC-8 total ambient 550 nm light scattering and a parallel dry light scattering for fine particles
(<1 μm). For context also included on Figure 5(c) is the inferred light extinction derived from the
UW HSRL by assuming a lidar ratio of 55 sr$^{-1}$. The period of averaging for the HSRL data is
19:00-21:00 UTC, or essentially from the start of the profile until just before the storms passed
overhead. Total particle counts from the LAS and CN counters are plotted on Fig 5(d). To
prevent any possible cloud water or precipitation shattering effects on the aerosol instruments,
CN, nephelometer, and LAS data was heavily cloud screened with data points removed for one
second before the arrival and two seconds after the exit of any cloud with LWC>0.005 g m$^{-3}$.
Finally University of Colorado aerosol mass spectrometer organic material and sulfate is
provided in Fig. 5(e). Only under very heavy ice content conditions does AMS data need to be
expunged from the profile. To reduce noise, a 5 second boxcar average was applied to the
particle counter and AMS data. Also to improve readability of PBL features, similar plots from
0-4 km are likewise included as Figures 5(f)-(j) respectively.
The DC-8 profile depicts intricate layering behavior throughout the free troposphere in a fashion
consistent with the UW HSRL backscatter. As expected, the temperature profile is largely moist
adiabatic ~ 6$^{\circ}$ C km$^{-1}$, indicating an atmosphere that has been modified by convective processes.
Moist layers, well depicted in the dewpoint sounding when it converges with temperature, often
coincided with minor temperature inversions. For reference these layers associated with
dewpoint depressions <2 $^{\circ}$C are labeled on Fig. 6 as lines, or for three deeper layers, shaded
bands. Characteristics of these layers are also provided in Table 1, and Appendix A.2 provides
images taken from the DC-8's forward video to provide visual context of the environment being
sampled. As expected, moist layers coincided with increases in $\omega_v$. However these layers also
strongly coincided with increases in other tracer species such as CO and dry aerosol
concentration. In the following subsection, we provide a narrative starting with layers influenced
by PBL detrainment (PBL layers 1 and 2; Sec. 5.1) followed by upper free troposphere



detrainment by the Cb (UT layers 1-4; Sec. 5.2). Emphasis will then be placed on the nature of
aerosol and Ac layers in the middle free troposphere (MT Layers 1-3; Sec. 5.3). Finally we will
examine composition and particle properties between these layers (Sec. 5.4).
5.1 PBL Detrainment Layers
Our first area of examination is of detraining aerosol layers associated with the development of
the PBL, with clouds ranging from CuHu to Cu and the occasional congestus. This baseline PBL
environment is described in detail in Reid et al., (2017), and is the subject of a subsequent paper
on particle transformation and inhomogeneity within the PBL. Here, we consider a few specific
aspects of the DC-8 data set to aid in overall profile interpretation, and also in the analysis of
covariability among aerosol, water vapor and Ac cloud formation in the middle troposphere.
To begin we examine the nature of the PBL's mixed layer as this is the "source" of the
atmospheric constituents being convectively lofted. However, the observation of the PBL's
mixed layer profile at the bottom of the profile is contrary to what one would expect. Most
notably, the $\omega_v$ is not constant with height near the bottom of the profile, suggesting that either
the environment is not well mixed or the DC-8 never made it into the mixed layer. Based on
forward video (Fig A.3 (a)), the spiral was initiated below cloud base and there was a strong
gradient in $\omega_v$ on approach to the spiral; in fact isolated showers were seen across the horizon. It
is therefore likely that the mixed layer is influenced by regional gradients- a recurring problem
with profiling with large and fast moving research aircraft. Likewise, at the start of the spiral,
gradients are also detected in CO and aerosol variables. These gradients are good indicators of
significant spatial variability of atmospheric constituents in the mixed layer. Using a single point
at the top of the mixed layer just before ascent as a baseline (Table 1), $\omega_v$ and CO were at a
maximum of the profile at 15.5 g kg$^{-1}$ and 110 ppbv, respectably. CN was at 2300 cm$^{-3}$, and a
LAS volume concentration of 2.8 μm$^3$ cm$^{-3}$ for an index of refraction of polystyrene spheres,
(n=1.55), consistent with AMS concentration of particulate organic matter and sulfate of 4.2 and
1.5 μg m$^{-3}$, respectively. The light scattering hygroscopicity of growth from 20-80% RH was
1.62, typical of the region (Wonaschuetz et al., 2012).
Within the nearest level to the surface (PBL Layer 1 in Fig. 6, ~1.6 km MSL, 1.4 km AGL) is a
clear aerosol enhancement just at and above mixed layer top which we diagnosed at ~1.5 km
through a combination water vapor and temperature and visual inspection of cloud base from the
forward video. An enhancement is expected in ambient scattering at the top of the mixed layer
due to the increases in humidity with height in the mixed layer coupled with aerosol
hygroscopicity. But just above the mixed layer there is an increase in CO, dry aerosol mass,
number, CN and scattering. This, like the mixed layer variables, might be an aliased signal, but
also is influenced by detrainment from the Cu clearly present (Figure A.3(b)). At Huntsville at
the same time as the DC-8 spiral, the unaliased HSRL profile showed classic increased aerosol
backscatter (and presumed extinction) to a maximum at a level of 2 km MSL, indicating the top
of the mixed layer and cloud base slightly higher than the spiral location. PBL layer 1 is made up
of consecutive spikes within $\omega_v$, CO, dry light scattering, LAS and CN concentrations, and AMS
sulfate as the DC-8 passed through the top of the mixed layer and into the level of the lowest
cloud bases (~1.5km AGL; Fig. A.3 (b)). Dramatic increases in CN and sulfate in particular
suggest that this layer potentially hosted secondary particle mass production via detrainment
from nearby shallow clouds (e.g., Wonashuetz et al., 2012). Although RH values were on the
order of 85-90%, both the probe data and visual inspection of the video data show this peak is
not associated with any form of cloud contamination. Ultimately, evidence suggests that this
layer is detrainment of mixed layer air from small cumulus. Even though this location near the
Tennessee River hosts some sporadic industry on its shores, the nature of the tracers, such as
water vapor and CO, demonstrates this layer was convectively transported from above the mixed
layer by small Cu. Recent studies suggest that the oxidation of $SO_2$ to $SO_4^=$ in such clouds can be
extremely fast (e.g., Loughner et al., 2011; Eck et al, 2014; Wang et al., 2016).
The second layer analyzed, PBL layer 2, was much deeper than the first, at 2.5-3.2 MSL (Fig.
A.3 (c)). This layer can be classified as the upper portion of the PBL entrainment zone, where air
is actively mixing with the free troposphere above via detrainment from cumulus. The $\omega_v$ is
enhanced and, between clouds, relative humidity ranged from 80-90%. At times enhancements
existed in LAS particle number and in AMS sulfate and OC. Spikes in CN concentration reached
10,000 $cm^{-3}$, likely a product of convective boundary layer precursor emissions receiving high
actinic flux not only directly from the sun, but also reflected from nearby clouds (e.g., Radke and
Hobbs, 1991; Perry and Hobbs, 1994; Clarke et al., 1998). Also during the DC-8's transit of this
layer was a 15 second Cu penetration that included significant precipitation, although this period
is expunged from the aerosol particle counter record in Fig. 6. In the middle of this cloud, CO
reached 80 ppbv, indicating convective lofting of mixed layer air. It is this cloud that we believe



developed into the CB sampled. At the time of this first penetration, from visual inspection, the
cloud top could not have been more than ~1 km above the aircraft (Fig A.2 (c)), consistent with it
not being picked up with NEXRAD.
The PBL Layer 2 detrainment environment is discussed in detail in Reid et al. (2017), and owing
to convective pumping and cloud processing of mixed layer air and high relative humidity
contributes significantly to regional AOD variability. Sometimes described as cloud halo effects
to explain covariability in cloud fraction and AOD, this PBL Layer 2 is actually a wide spread
detrainment induced layer (Reid et al., 2017). This layer was visible not only on the DC-8
nephelometer and AMS data, but is also coincident with a strong aerosol return from the
Huntsville lidar, some ~100 km to the west. Notably, the top of this layer coincides with the
lifting aerosol layer topped by Ac clouds in UW-HSRL (Fig 1(f), Fig 3(a) and Fig 4(c)) and
serves as a potential boundary between the PBL and free troposphere.
5.2 Upper Free Troposphere
Moving from PBL influenced aerosol layers, we now briefly examine the region dominated by
convective outflow from the anvil, diagnosed as detrainment in association with ice. This altitude
domain is largely outside the scope of this paper, and will be discussed in detail in other
SEAC$^4$RS papers. Nevertheless, for completeness a brief description is provided here. Like the
top of the PBL, the bottom of the cirrus anvil outflow layer is ambiguous. From Fig. 4 and in
particular Fig 5(a), it is clear that liquid water could exist as high as 8.5 km, or ~ -21 $^{\circ}$C,
although ice was clearly nucleating and falling below this liquid water. The first full ice layers
were experienced by the DC-8 at 8 km and 8.4 km  (UT 1 and 2, Fig A.3 (g) and (h)) followed by
a second cirrus cloud (UT2) a third at 9.4 km (UT3; Fig A.3 (i)), and finally a deep cirrus
penetration from 10-11 km (UT4; Fig A.3(j)). Because cloud particles in these layers were
entirely made of ice, with ice water content approaching 1 g m$^{-3}$, aerosol size and scattering data
are not available; although prominent peaks in CO, sulfate, and particulate organic matter are
found at each level indicating convective pumping and detrainment. From an aerosol point of
view, it is obvious that significant enhancements in particle mass and number exist on either side
of the cirrus layer. Notably the boundaries of these layers were enriched in organics relative to
sulfates, and CN>10 nm concentrations were on the order of 10,000-20,000 cm$^{-3}$, particularly
above 9.5 km. Indeed, observations suggest that deep convection is highly efficient at
transporting boundary layer air through to the anvil (Yang et al., 2015).





### 5.3 Middle Free Tropospheric Layers

The focus of this paper is on the middle tropospheric detriment layers, bounded below by the primary PBL detrainment layer and its associated Ac clouds and above by the anvil cirrus, both described above. Within the middle troposphere there were numerous perturbations in water vapor, CO, and aerosol features. In particular, three coincident water vapor, CO and aerosol layers were observed in the DC-8 spiral, clearly associated with liquid water clouds (MT Layers 1, 2, and 3; Fig A.2 ((d), (e), (f)). Starting from the bottom of the free troposphere and working upwards, a slight inversion at 4.1 km delineated a rather minor water vapor and aerosol layer (Fig. 6 MT1; Fig. A.2 (d)), which, like Layer PBL2, spanned both the DC-8 profile and the UW-HSRL lidar at Huntsville. The inversion associated with this layer was a 200 m deep area having a near constant temperature of $3.4^{o}$C. Visual inspection of video data suggests this level was associated with the maximum heights of the larger Cu and likely represents the very top of convective pumping by larger boundary-layer clouds (Fig. A.3(d)). Such an interpretation is also consistent with this layer delineating a drop in aerosol light scattering and mass which has likely detrained from these larger clouds. Yet coincident with this inversion is a small spike in particle number, as measured by the CN counter. The similarity of this layer to PBL2 is noticeable, even if ejections are more sporadic than the smaller and more numerous cumulus clouds in the region that define PBL2. These layers  may be isolated, or be associated with a more organized region, but they nevertheless show the lofting of mixed layer air into the free troposphere. Indeed, this layer reminds us that in convective environments the physical top of the PBL is difficult to define; the boundary between the cloud tops and the free troposphere is variable.

Special attention is paid here to the next two layers (MT 2 and MT 3) where significant perturbations to tracer and aerosol loadings were associated with thin Ac cloud decks. Within MT2, a strong aerosol return was present at 4.6 km associated with a shelf cloud deck at $\sim0.5^{o}$C detraining from the sampled Cb (Fig. 2(d) and Fig. A.3 (e)). MT3 contained a deeper layer of isolated Ac clouds from ~6 to 7 km (-6 to $-12^{o}$C; Fig. A.3(f)). Unlike layers below these, they are not directly observed at Huntsville, but are similar to a case earlier in the day and after the storm passes later in the day (e.g., Fig. 5(b) and (d)). Detailed timeseries of data as the DC-8 passed through these two layers are presented in Figure 7.

MT2 at 4.6 km was targeted for direct penetration by the DC-8 because it represented a classic melting level Ac detrainment shelf commonly observed around the middle of Cbs (e.g., Fig. 1(a);





Johnson et al., 1996; Posselt et al., 2008). The DC-8 approached the cloud from the side at a
slow climb rate (~ 1 m s$^{-1}$), and flattened out for Ac cloud sampling, followed by a more
accelerated climb (Fig. 7(a)). Consequently, the DC-8 captured the environment below and to the
side of the Ac deck, and the Ac deck itself. Given the air speed of ~156 m s$^{-1}$, the 50 second
timeseries for this aerosol and cloud layer spans ~8 km. On approach, water vapor, CO, dry light
scattering and aerosol mass species also increased in a layer perhaps only 200 m thick. Water
vapor changed in a series of steps, suggesting coherent layers, including a very sharp drop in
water vapor for only a few seconds just before cloud penetration, only to drop again on exit. The
drop in $\omega_v$ and cloud liquid water was immediately below a 2$^o$C magnitude temperature
inversion.
Aerosol particle counts for $d_p>0.1\mu$m (and particle volume, not shown) also increased on
approach to the Ac. Total CN ($d_P>10$ nm) however dropped precipitously suggesting an overall
shift in the background size distribution in an environment that disfavored nucleation. Cloud
penetration lasted ~20 seconds (~3 km) and cloud liquid water contents ranged from 0.12 to 0.18
g m$^{-3}$. Droplet effective radius from the cloud probes (not shown) was consistently in the 4.5-6
$\mu$m range.  Not surprisingly with a cloud temperature of ~1$^o$C no ice was present.  While aerosol
number or size distributions are unavailable during cloud sampling due to inlet shattering, CO
clearly peaked within 200 m of the altitude of the cloud. Yet, the AMS showed a decrease not
only within the cloud, but also just before cloud entry. As the DC-8 climbed up and away from
the Ac deck, LAS particle counts and AMS OC and sulfate dropped, while CN returned to
baseline levels and even spiked for a short period. Overall, MT2 observations match qualitatively
what was seen in the HSRL data, with the cloud resting on the top of the aerosol layer.
While MT2 was associated with a thin detrainment shelf, Layer MT3 was representative of a
much deeper layer of convective detrainment, spanning the 6-7 km level. These layers can be
visualized in the Huntsville HSRL data in Fig. 5 (b) and (d). Sampling of this layer was in the
form of steps (Fig 7(g)). Throughout this layer, $\omega_v$ and relative humidity varied in such a way
that this overarching layer is most likely an agglomerate of many layers. The existence of several
thin layers at various heights may result from detrainment at the tops of terminal congestus with
termina at different levels (Moser and Lasher-Trapp, 2017). Consequently, very faint Ac clouds
were visible on the video (e.g., Fig. A.3. (f)), though there were few actually cloud penetrations.





The clouds sampled had very meager liquid water contents (<0.01 g m$^{-3}$); barely clouds. Yet,
these clouds were mixed phase with ice clearly visible in 2D probe data at temperatures of -10$^{o}$
C, (Figure A.4; annuluses are also ice out of focus).  Such ice is not noticeable in lidar data, as
optics may be still dominated by spherical liquid droplets. Thus, this observations suggests that
low depolarization observations cannot exclude the presence of ice,
For most of MT3, $\omega_v$ and CO varied in concert. However, at the very top of the level, they
quickly become anti-correlated-suggesting water vapor at this location is not being brought from
the boundary layer. Instead, it may be from entrained air along the sides-perhaps along cloud
edges air entraining in is the first to detrain out (Yeo and Romps, 2013). Aerosol data is not
much more enlightening. Aerosol mass was rather steady, and at reduced concentration than its
lower level counterparts. At the same time, spikes in aerosol counter and nephelometer data
occurred near clouds, and may just as easily be a result of droplet shattering artifact rather than
convective pumping.
5.4 Vertical Profile Aerosol Chemistry and Mass
5.4.1 H$_2$0 and CO
Previous subsections in Section 5 describe the nature of individual detrainment layers. In this
final subsection, we provide a closer examination of differences in their properties. If we
conceptualize the environment as being influenced by shallow to deep injections of mixed layer
air being convectively transported to the free troposphere by clouds entraining and detraining air
along the way, it is best to start with reliable tracers such as CO. Figure 8 includes profiles of the
ratio of aerosol number and mass to excess CO.
Paramount to all subsequent interpretation of the profile is the molar ratio of excess water vapor
to CO. Whereas we can take background CO value of 60 ppbv (or any nearby value as long as
we are consistent), water vapor is a bit more problematic. We derived excess water vapor by
taking advantage of the deep convection horizontal scope of several hundred kilometers upwind
of UAH. A background value was derived from the average mixed layer mixing ratio, followed
by a 4$^{th}$ order polynomial fit against pressure above (r$^2$=0.99) . The calculated excess $\omega_{\underline{v}}$ between
the DC-8 and UAH sounding is provided in Fig. 8(a). As expected, $\omega_v$ is enhanced in the vicinity
of convection, notably in the mixed layer, as well as individual PBL and mid-level detrainment





layers, such as 3 km (PBL2), 4.6 km (MT2, 0°C), 6-7 km (MT3). Water vapor is also more
broadly enhanced in the upper troposphere layers (UT1-4).
Moving from establishing the background water vapor profile, we next consider how a parcel of
air lofted into the PBL deviates from textbook descriptions during deep convection. If the parcel
ascends without mixing, the water vapor mixing ratio is expected to decrease with altitude, as
temperature decreases at the moist adiabatic lapse rate and water vapor is removed by
condensation and precipitation. In contrast, CO is expected to remain constant over the time
scale of convective ascent. In reality, the vertical profiles of both constituents are modified by
entrainment/detrainment processes, and theory and numerical experiments indicate there are few
truly undiluted parcels to be found anywhere in regions of shallow or deep convection (Zipser
2003; Romps, 2010; Romps and Kuang, 2010). Parcels that ascend in a region near the core of
convection (far from the cloud edge) may conserve CO and approximately follow a moist
adiabat. Parcels closer to the cloud top and edge will undergo mixing with air that has originated
from various levels inside and outside of the cloud, and may reflect multiple entrainment-
detrainment events (Yeo and Romps, 2013). The ratio of water vapor to CO concentration in
undiluted ascent should be uniquely determined by the parcel's initial properties in the mixed
layer, and departures from this ratio within the cloud reflect the action of mixing. Outside of the
cloud, the situation is a bit more complicated. We expect water vapor content to decrease with
height, and, if CO is well mixed, then the concentration will be constant with height. Increases in
the ratio of water vapor to CO with height reflect the action of detrainment from convection, as
water vapor decreases with height more rapidly than CO.
The 0°C melting level is further related to the the air parcel characteristics. The molar profile of
excess $H_2O$ to CO ratio is provided in Figure 8(b), and throughout the lower troposphere the
ratio increased to a maximum at the 0°C melting level. This increase reflects a more rapid
decrease in CO with height relative to water vapor, and is punctuated by two local maxima in the
ratio at 1.5 km and 3 km above the surface. Above the melting level, the ratio of $H_2O$ to CO
precipitously drops, then exhibits local maxima at 5 km and 5.5 km.
Examining possible causes of the water vapor and CO ratio variability in the vertical above the
0°C melting level entails a closer examination of the impacts of detrainment on an air parcel.
Detrainment of air from convection results in local increases in both water vapor and CO;





however, water vapor content in detrained air will be greater than CO due to evaporation of
cloud condensate. The general increase in water vapor to CO ratio indicates the repeated action
of entrainment/detrainment (and evaporation of cloud condensate) around developing cumulus
clouds, while local maxima in water vapor to CO ratio reflect the action of enhanced detrainment
at specific levels; in this case, the tops of CuHu and Cu at 1.5 and 3 km, respectively. Detraining
air from congestus and deep convection at the melting level provides the strongest local source
of water vapor (direct and via evaporated cloud), and also the largest water vapor to CO ratio.
Contrary to the spikes in water vapor content caused by detrainment, immediately above the
melting layer, water vapor content is very low as this air originates in the middle and upper free
troposphere (c.f., Figs. 4 and Posselt et al. 2008). CO contently remains relatively high, since CO
is relatively well mixed in the middle and upper free troposphere (Fig. 5b). The near
discontinuity in water vapor content in the vertical, coupled with relatively small changes in CO,
result in the rapid decrease in water vapor to CO ratio above the melting layer. Relatively high
CO concentrations in the air detrained at and below the melting layer can be seen in the profile of
CO (Fig. 5b) and in the aerosol number to CO ratio maxima in Fig. 7b. Above the melting layer,
such as in the 6-7 km region (MT #3) thin layers of high water vapor to CO ratio are likely due to
detrainment from cumulus congestus clouds.
**5.4.2 Aerosol Mass**
Moving to aerosol particle profiles, different aspects of convective transport reveal themselves.
The ratio of LAS particle concentration ($d_p$>0.1 μm, representing the accumulation mode) and
CN ($d_p$>10 nm, representing the nucleation mode) to CO is presented in Figure 8(c). Relative to
CO, accumulation mode particles largely drop continuously in number from the surface to $0^oC$
level. Positive perturbations exist within the PBL and MFT aerosol layers as diagnosed in Fig. 6.
At heights above the $0^oC$ level, the accumulation mode to CO ratio stabilizes at lower
concentrations with occasional layers. There is some difference in light scattering (Fig. 8(d)) and
OC and sulfate from the AMS ((Fig. 8e)), where we find mass enhancement in the PBL
detrainment zone.
Nucleation mode aerosol becomes more prominent with height owing to more intense solar
radiation and a decrease in available accumulation mode surface area. Nucleation rates of



particles from precursors detrainment from anvils can be rapid (Waddicor et al., 2012).
Detrainment layers host strong positive and negative perturbations in CN count, which does not
project significantly onto light scattering or mass, inverse with the concentration of accumulation
mode particles which do project strongly onto optical observables.
To explore variability in particle size distributions in the vertical, Fig. 9(a) and (b) provides LAS
number and volume size distributions for key levels throughout the profile, and is consistent with
what can be inferred from Figure 8. Best fit baseline particle size distribution within the mixed
layer suggest Count Median Diameter (CMD) and Volume Median Diameter (VMD) of 0.14 and
0.25 µm, respectively. At the first layer (PBL 1), dry particle size CMD and VMD increases to
0.16 and 0.28 µm, respectively, at the same time of increases in particle mass relative to CO.
This is all consistent with secondary aerosol particle mass production on exiting particles. After
this point, we find a reversal in particle CMD and VMD with height. This is suggestive of
precipitation scavenging of larger particles in larger clouds the deeper the detrainment. That is,
particles that are detraining from smaller non-precipitating clouds keep their secondary produced
mass. However these same aerosol particles that enter deeper precipitating clouds not only lose
their larger particles due to droplet nucleation, but also the recently gained secondary mass.
Nevertheless, significant aerosol mass from the boundary layer still be ejected in the anvil  as
evidence in the 9-11 km altitude range in Figs 8 (d) and (e).
**5.4.3 OC and Sulfate**
The $0^{\circ}C$ level is clearly a delineator in the sulfate to OC ratio (Fig. 8(f)). Near the surface the
ratio of sulfate to OC is ~0.4. In the first PBL detrainment layer (PBL1) there is a doubling of
sulfate relative to CO. Such a mass increase relative to CO may be indicative of secondary
aerosol production-and indeed sulfate peaks in this layer not only against CO, but also relative to
OC (Fig. 7(e, f)). Particulate organic matter mass relative to CO peaks in PBL Layer 2, but with a
reduction in sulfate. Detrainment from this layer is associated with deeper clouds, including
warm precipitating clouds in the immediate vicinity. Thus, sulfate particles may be preferentially
scavenged.
The ratio of sulfate to OC further changes systematically through the profile, decreasing to a
minimum just below 4 km. This, coupled with the decrease in accumulation mode number





relative to CO, may be a further indicator of aerosol particle processing and scavenging in
clouds. Above 4 km, sulfate increases again, perhaps due to oxidation of residual interstitial or
dissolved but on oxidized sulfur species in either Ac clouds or in gas phase. This increase may
also be related to the relative mass distribution within detraining cloud droplets. Sulfate mass
fractions do appear to recover in the upper troposphere, perhaps due to homogenous nucleation
of the small amount of $SO_2$ detraining from sublimating ice.
**6.0     Discussion-combining datasets and hypothesis development**
The purpose of this paper is to demonstrate on a canonical day that aerosol layering
characteristics in the free troposphere and PBL entrainment zone are delineated by cloud
structure and its associated thermodynamic profile. Examination of this day leads to many
questions about aerosol processes and potential impacts or feedbacks with understudied Ac
clouds.   In the following section we use the combined datasets from the UW HSRL and the DC-
8 aircraft to formulate several hypotheses about Ac formation that need further attention by the
community.
6.1 Hypothesis: Ac cloud's low liquid water and slow updraft velocities are susceptible to small
changes in the CCN population:
One of the most remarkable aspects of next generation lidar systems such as the UW HSRL used
here and new Raman systems such as described in Schmidt et al., (2015) is their ability to
observe intricate aerosol features at very low particle concentrations. Fig. 3(d), 4 and 5
demonstrate fine coherent structure of aerosol layers in the free troposphere that, in the past,
were rarely quantified. Even with aerosol backscatter levels at or even under $<5\text{x}10^{-8}$ (m sr)$^{-1}$, or
<5% of Rayleigh backscatter, aerosol layers of only a 100 to a few 100 meters thickness are
clearly visible, and can persist for hours undergoing gravity wave undulations along with gradual
changes in observed layer height at the meso to synoptic scales. Ac are often associated with
observed aerosol layers, and the clouds we observed had very low liquid water contents of a few
tenths of a g m$^{-3}$ at most (e.g., Fig. 7). Drawing from parallels to Stratocumulus (e.g., Martin et
al., 1994; Platnick and Twomey 1994; Ackerman et al., 1995), or the very limited available
measurements of such relationships for Ac in the field (e.g., Reid et al., 1999; Sassen and Wang,
2008; Schmidt et al., 2015), we would expect Ac cloud's low liquid water and slow updraft
velocities to have significant sensitivity to CCN populations. It stands to reason that aerosol-Ac



sensitivities can then project onto cloud reflectivity, cloud lifetime and consequently the local
energy budget.  Thus trends, in global aerosol populations, that regionally have strongly varying
signal and sign, (e.g. Alfaro-Contreras, et al., 2017) may very well result in large scale trends in
Ac cloud cover (e.g., hypotheses by Parungo et al., 1994) or reflectivity. However estimating
CCN concentration based on the regional aerosol loading is difficult.  One is attempting to
estimate the properties of a very thin aerosol layer with high complex relationships to the
boundary layer and regional convection.
Given the difficulties in modeling aerosol entrainment and entrainment processes, one might
think that direct observation would be much more straightforward.  But the convection-Ac
system is very difficult to monitor. Despite improvements to lidar systems, data from lidars are
underdetermined. Aerosol backscatter and/or extinction, even spectrally resolved, are only semi-
quantitatively related to CCN concentrations. To provide aerosol microphysics information, an
aircraft such as the DC-8 is required. But in the context of the aerosol structure highlighted in Fig
5, aircraft sampling is hopelessly aliased. This is compounded by the typical structure of a thin
Ac deck above its associated thin aerosol layer. Broad sampling of the free troposphere would
reveal only period collinear perturbations, and aircraft location relative to the rest of the fine
aerosol structures would remain unknown. Even if the DC-8 were directly over the Huntsville
site, interpretation of the data would be complicated by features such as gravity waves and halos
around individual clouds. Therefore, much effort is required on sampling methods to address this
hypothesis.
6.2 Hypothesis: CN events can sustain and enhance CCN populations in Ac clouds
The impact of aerosol dynamics of the region must be considered when addressing a number of
science questions. Aerosol backscatter is dominated by accumulation mode particles that, owing
to their size, also make the best CCN. While there are copious CN, there are few particles in
number of any appropriate size to behave as CCN (~100 cm$^{-3}$ or less in the LAS at altitudes
above the 0$^{\circ}$C level). Considering the proclivity of CN nucleation events, and the overall
increasing numbers of CN at higher altitudes, the CCN versus optical detection relationship is
complex, (e.g. Schmidt et al., 2015). Enhancements in accumulation mode particles near Ac
appear to be anti-correlated with CN for this case-likely due to available surface area for
secondary mass production and or coagulation. At the same time, explosive nucleation events are



visible and expected. This all leads to questions about layer flow dynamics in and around Ac and
their associated aerosol layers and/or halos. Does the cycling of air through an Ac feedback into
its own CCN budget? Does non-precipitating cycling enhance particle size and hence CCN
number for any given supersaturation? In precipitating Ac, where are replacement CCN coming
from, and do nucleating CN ever offer a supply? Or, as a hypothesis, perhaps CN events can
sustain and enhance CCN populations in Ac clouds. The null hypothesis would then be that CN
are consumed in individual droplets and have little overall effect in clouds with such meager
updraft velocities and super saturations. This topic in particular needs to be addressed in highly
detailed modeling studies.
6.3 Hypothesis: At and below the melting level, air is dominated by detrainment of boundary
layer air and above the melting level in the middle free troposphere, air is more influenced by
entrainment and detrainment along the cloud edges. However PBL air can be ejected through the
anvil.
This hypothesis or ones like it is related to the fundamental "plumbing" of convection and what
fraction of air from which levels is transported where. Much of the combined Ac/aerosol
environment rests on the nature of convective detrainment and this detrainment phenomenon
may give insight into cloud dynamics and transport. The updraft core is somewhat insulated from
entrainment/detrainment processes, whereas parcels closer to the cloud top and edge will
undergo mixing with air that has originated from various levels inside and outside of the cloud.
Observations around clouds may reflect multiple entrainment-detrainment events (e.g., Yeo and
Romps, 2013). The ratio of water vapor to CO concentration in undiluted ascent should be
uniquely determined by the parcel's initial properties in the mixed layer, and departures from this
ratio within the cloud reflect the action of mixing. Detraining air from deep convection at the
melting level provides the strongest local source of water vapor (direct and via evaporated
cloud), and also the largest water vapor to CO ratio. We hypothesize that, up to the melting level,
detrainment is dominated by boundary layer air, whereas above this level air is more influenced
by entrainment and detrainment along the cloud tops and edges. It is noteworthy also that the Ac
cloud observed on the DC-8 was not directly at $0^{\circ}$C, but rather $0.75^{\circ}$C, consistent with the
formation of an inversion directly above it (e.g., Figure 7(b), T minimum not exactly at $0^{\circ}$C, but
rather at $0.5^{\circ}$C). These observations are in agreement with the simulations by Posselt et al.,





(2008) and Yasunaga et al. (2008), both of which were modeling studies that managed to form
melting level clouds without any predefined environmental area of stability. Perturbations in
temperature may be representative of large scale vertical motions on the outside of the clouds,
including downdrafts adjacent to regions of in-cloud upward motion.  Schmidt et al., (2014)
suggested that the heating/cooling differentials in the vicinity of altocumulus clouds can result in
areas of mesoscale subsidence, further perturbing flow fields and presumably CCN intake into
these clouds.
We leave open the possibility that depending on storm dynamics, parcels in the inner core of
convection can be ejected into, and out of, the anvil. This overall structure, with PBL air at cloud
tops and bottoms, with more entrainment/detrainment dominated properties is supported in figure
8 where aerosol mass ratios to CO are given as well as an altitude dependence of sulfate to
organic matter is given. So clearly different altitude ranges have strong relationships to cloud
entrainment and detrainment processes and the overall convective structure. Models can certainly
provide insight, but considerable thought must be given to verification.
**7.0    Conclusions**
This paper presents August 12, 2013 as a case study from the SEAC[4]RS campaign that
demonstrates Altocumulus cloud (Ac), aerosol and water vapor layering phenomena in a
convective regime over the southeastern United States (SEUS). This day was chosen due to
proximity of the DC-8 research aircraft to a High Spectral Resolution Lidar (HSRL) at
Huntsville Al. The HSRL gives period level perspective on Ac clouds and their observed aerosol
"halo" to help interpret in situ DC-8 data.  Analysis of the meteorology of the region on this day
supported the assertion that aerosol was "local" to the SEUS and thus should be considered to be
representative of regionally forced convective environments. A 33 hour sample of lidar data was
presented to demonstrate the diurnal cycle of cloud and aerosol features in this convective
environment. The HSRL provided aerosol backscatter and precisions at or better than 5% of
Rayleigh, and demonstrated extraordinarily fine aerosol features in the vicinity of altocumulus
clouds formed in the outflow of deep convection. This day was in turn compared to a DC-8
profile conducted that afternoon on the downwind side of a developing storm providing in situ
data on the middle free troposphere aerosol environment.





Aside from typical boundary layer development and cirrus outflow, numerous aerosol and Ac
decks were identified, many of these Ac produced ice virga. Ac formed at the top of the residual
of the previous day's planetary boundary layer entrainment zone, where air was largely
influenced by boundary layer cloud detrainment. This layer formed in the morning hours, and
increased in base altitude with the developing boundary layer below it. Such rising may be a
result of mesoscale flows or cloud lofting.
Above the PBL-top Ac, several other combined aerosol-Ac-water vapor layers were observed.
Including 1) a 4 km detrainment layer that we surmise is from the very tops of cumulus
mediocris clouds; 2) layers at or just below 4.7 km/0°C melting level representing deep
convective detrainment shelves, and 3) 6-7 km layers consistent with that appear to be consistent
with detrainment from the tops of congestus clouds. From the HSRL, Ac clouds were associated
with clear aerosol "halos", typically with Ac clouds on top. The intensity of aerosol backscatter
associated with Ac cloud halos appeared to decrease with height, beyond what would be
expected from adiabatic expansion. The lowest Ac clouds associated with PBL entrainment zone
have larger returns associated with their proximity to the polluted PBL and large accumulation
particle size, and hygroscopicity. However, middle free troposphere layers had markedly smaller
accumulation mode sizes with height, but higher CN counts. Aerosol layers above 0°C had the
smaller accumulation model sizes and highest CN concentrations. This is consistent with further
cloud processing and scrubbing of detraining air at higher altitudes. Particle size and composition
data suggest that detraining particles undergo aqueous phase or microphysical transformations,
while at the same time larger particles are being scavenged.
Examination of profiles suggest an excess of water vapor and aerosol particles relative to CO
within and above the PBL entrainment zone to the melting level, and observations around clouds
may reflect multiple entrainment-detrainment events (e.g., Yeo and Romps, 2013). We expected
the ratio of water vapor to CO concentration in undiluted ascent should be uniquely determined
by the parcel initial properties in the mixed layer, and departures from this ratio within the cloud
reflect the action of mixing. Detraining air from deep convection at the melting level provides
the strongest local source of water vapor (direct and via evaporated cloud), and also the largest
water vapor to CO ratio. We hypothesize that up to the melting level, detrainment is dominated
by boundary layer air, whereas above this level air is more mixed involving




entrainment/detrainment along the clouds. Water vapor flux to the middle free troposphere may
also be enhanced by evaporating precipitation, whereas higher altitude parcels undergo
dehydration.
This work leads to numerous questions regarding relationships between aerosol layers and the
properties of Ac clouds. It has been long hypothesized that increasing trends in aerosol
concentrations over the past decades will result in more convective lofting, and then perhaps an
indirect effect in associated Ac clouds and perhaps increases in cloud lifetimes (e.g., Parungno et
al., 1994). The observation that Ac clouds have visible halos of accumulation mode particles
certainly indicates that Ac are coupled with the boundary layer aerosol system. Enhancements in
accumulation mode particles near Ac appear to be anti-correlated with CN for this case-likely
due to available surface area for secondary mass production and or coagulation. At the same
time, explosive nucleation events are visible and expected in the vicinity of clouds. All of this
suggests complex CCN-Ac coupling and questions about layer flow dynamics in and around Ac
and their associated aerosol layers and/or halos. Does the cycling of air through an Ac feedback
into its own CCN budget? Does non-precipitating cycling enhance particle size and hence CCN
number for any given supersaturation? In precipitating Ac, where are replacement CCN coming
from, and do nucleating CN ever offer a supply? Or, as a hypothesis, perhaps CN events can
sustain and enhance CCN populations in Ac clouds. The null hypothesis would then be that CN
are consumed in individual droplets and have little overall effect in clouds with such meager
updraft velocities and super saturations.

**8.0 Author contributions**
JR: Lead author and investigation; DP, KK, & RH: investigation and manuscript composition;
ST, CT, SW, & LZ: Flight, data, and science support; All others data providers

**9.0 Acknowledgements.**
We are grateful to NASA Atmospheric Composition Focus Area for their sponsorship of the
SEAC[4]RS campaign, as well as to all of the senior leadership, management and scientists that
contributed to this successful mission. Funding for the deployment of the UW-HSRL was
provided by the CALIPSO science team as a contribution to the SEAC[4]RS program. Analysis of
the data presented here was provided by a NASA Atmospheric Composition Campaign Data
Analysis and Modeling program (NNH14AY68I) and the office of Naval Research Code 322
(N0001414AF00002). The SEACIONS network, organized at by PI Anne M. Thompson and



Jacquie Witte at NASA/Goddard NASA/Goddard was initially supported through a grant to
Pennsylvania State University (NASA NNX12AF05G). PCJ and JLJ acknowledge support from
NASA NNX15AT96G. We are grateful to SPEC incorporated (esp. Paul Lawson) for providing
cloud probe data, and Jose Jimenez (University of Colorado) for providing aerosol mass
spectrometer data. A portion of this research was carried out at the Jet Propulsion Laboratory,
California Institute of Technology, under a contract with the National Aeronautics and Space
Administration. Airborne data doi:10.5067/Aircraft/SEAC4RS/Aerosol-TraceGas-Cloud. All
SEAC⁴RS DC-8 and geostationary data is available at https://www-
air.larc.nasa.gov/missions/seac4rs/. All HSRL lidar data used in this analysis is available at
http://lidar.ssec.wisc.edu/. MODIS satellite data used in this mission was downloaded from
ftp://ladsweb.nascom.nasa.gov/

**10.0    Appendix A. Supplemental meteorology analysis and imagery**

This appendix includes a meteorological analysis of August 12, 2013 and corresponding figures

to support the interpretation of this study. Included is Figure A.1 of NEXRAD reflectivity

spanning the study period, with higher temporal resolution when the DC-8 was sampling the

storm. Marked is the Huntsville site (red circle) and the location of the DC-8 aircraft. Figure A.2

provides GOES 13 11 μm channel images of the storm that produces Ac clouds in the Huntsville

lidar data in Fig. 5(d). (a) 12 Aug 2013, 1715z highlighting PBL detrained Ac clouds. Subsequent

panels show with an arrow the back trajectory location with corresponding cloud top

temperatures: A.2 (b), Initiation time for the back trajectory to the $0^{o}$C cloud. (c) 10 hour back

trajectory endpoint to large detrainment shelf (d) Cb that formed the AC layer.  Tracking his

observed layer suggests it was transported ~ 350 km. Figure A.3 provides images from the DC-8

forward video for different altitudes and layers along the DC-8 spiral.

To provide context to this analysis, we provide a meteorological overview of the region during

the early phases f the SEC4RS study. August 5-14, 2013 was a convectively active period over

the SEUS during the summer of 2013. Weak mid-level shortwaves or cold and stationary fronts

impinging on high pressure along Southern Mississippi, Alabama, and Georgia brought

convective activity throughout the northern SEUS and Tennessee Valley. While scattered

afternoon precipitation formed throughout the region, a stationary front on August 11th over

southern Kentucky produced more substantial cells with series of southeastward propagating

outflow boundaries, leading to subsequent convection over northern Alabama and Georgia

through the day. One such band of Cbs passed through Huntsville in the early evening on August

11. By August 12th, convective available potential energy (CAPE) reached >1800 J kg⁻¹ at



sounding sites in the SEUS, leading to scattered Cbs forming in the early morning hours over
Tennessee and southeastern Missouri, and propagating into northern Alabama as the day
progressed. A significant line of convection reached the northwestern corner of Alabama, at
18:00 UTC (where it was sampled by the DC-8 at ~19:00 UTC), and subsequent convection that
formed on the eastward propagating outflow boundary reached the UAH lidar site 6 hours later.
Regional aerosol loadings for August 12[th] were consistent with air masses staying within the
SEUS over the past several days. AERONET AOD registered a 550 nm AOD of 0.18 at
Huntsville in the morning, and Terra MODIS AODs at 550 nm were reported that morning at
0.27 in the vicinity of the CB sampled. At the surface, regional $PM_{2.5}$ stations were reporting
daily averaged mass concentrations of 5-10 μg m$^{-3}$ at CSN and SEARCH sites. Specifically at
Huntsville, CSN $PM_{2.5}$ ranged from 10-14 μg m$^{-3}$ at daybreak and morning hours, dropping to 5-
10 μg m$^{-3}$ in the afternoon. Global models (e.g., Session et al., 2015) suggested no significant
long range aerosol transport into the region aside from a pulse of African dust around August 8[th]
and 9[th], three days before the case day studied here. There was no indication of smoke from the
Western United States impacting the area. HYSPLIT trajectories spawned at Huntsville were
consistent with transport via westerly winds on that day, in an air mass isolated from more
pollution in the north. Two day back trajectories showed that the middle troposphere air never
deviated from northeastern Mississippi and northwestern Alabama. Specific trajectories for Ac
layers identified also show origins from storms within this region over 350 km away(Fig. A.2).
All analyses indicate air masses near the surface through the middle troposphere were regional to
the SEUS over the past two days, representative of more regional pollution imbedded in a
regional convective regime.
Satellite cloud temperatures and the NEXRAD returns demonstrate the textural changes in cloud
fields as the day progressed from widespread cloudiness to more isolated cells. Above the mixed
layer, the sounding was moist but cloud free, with minor inversions at 3.4 km (perhaps indicating
the top of the PBL), 4.6 (0$^o$C) and 6.2 km heights. Winds were near constant at 250$^o$ above the
mixed layer, and with steady increases to 12 m s$^{-1}$ at the 0$^o$C melting level at 4.6 km providing
only a modest amount of shear (Fig 3(c)). Based on the satellite imagery and NEXRAD, the
fetch of the air mass over northern Alabama was over mostly Cu to a few isolated but non-
precipitating TCu clouds. The CAPE derived from the UAH sounding was 1650 J kg$^{-1}$, slightly



lower than all of the operational soundings surrounding the site at 12:00 (including Birmingham
to the south at 1831 J kg$^{-1}$ and Nashville to the north at 1811 J kg$^{-1}$). This neutral state in a
convective regime is the midday backdrop against which investigations of clouds in the vicinity
of isolated cells is performed in Section 4. By late afternoon, the region was more convectively
developed, with larger but more scattered individual storms. The one observed by the DC-8
began developing at 19:00 UTC and was monitored until 20:00 UTC (the location of the DC-8 is
marked on Fig A.2 (e) and (f), although the exact precipitating cell monitored was not observable
by NEXRAD until 19:35 when the cloud top height grew to above 6 km)). The last NEXRAD
return for this cell was at 20:00 UTC.
As the day progressed, Cbs repeatedly reformed and then propagated eastward, with one cell in a
mature phase reaching UAH site at 23:00 UTC. This pattern of afternoon thunderstorms
persisted for several more days, when large scale subsidence began to develop behind a weak
front that passed through on August 14$^{th}$.
At most levels at temperatures below -9 C intermittent ice was observed on the SPEC probes
(Figure A.4)  The SPEC cloud particle probes indicate ice was observed beginning about 19:27
UTC, at temperatures near -9 $^o$C, ranging in size up to around 400-500 μm.  Ice is observed on
the subsequent climb to colder temperatures at 19:34 UTC (-10 $^o$C), extending to sizes on the
order of 1 mm.  Intermittent ice, like that observed by the 2D-Stereo particle probe and shown in
Fig A.4, is observed at subsequently colder temperatures.  The 2D-S (Lawson et al. 2006) is a 2-
dimensional stereo particle optical array probe that records the cross sectional image of particles
from 10 μm to a few mm in size with 10 μm resolution for determining particle size,
concentration, extinction, phase, and ice particle habit.

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



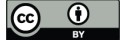




Table 1. Key physical attributes (mean and± standard deviation) of detrainment layers observed from the August.12, 2017 thunderstorm. Included are altitude in mean sea level (~300 m higher than above ground level), temperature and water vapor mixing ratio ($\omega_v$), carbon monoxide (CO), Laser Aerosol Spectrometer (LAS) number and volume at STP, 550 nm dry light scattering for particles less than 1 $\mu$m,  and Aerosol Mass Spectrometer (AMS) Organic Carbon (OC) and Sulfate at STP. Layers are defined as shown in Figure 6. [&]Mixed layer properties were taken as a 5 second average just before assent. *CO instrument was in a calibration cycle for part of this layer. #Upper troposphere

| | Altitude (MSL, km) | T (°C) | $\omega_v$ (g kg⁻¹) | CO (ppbv) | CN>10 (cm⁻³) | LAS N (cm⁻³) | LAS CMD/mode (μm) | LAS V (μm³ cm⁻³) | LAS VMD/mode (μm) | $\sigma_s$ 550 nm (Mm⁻¹) | $f$(80) | OC (μg m⁻³) | Sulfate (μg m⁻³) |
|---|---|---|---|---|---|---|---|---|---|---|---|---|---|
| ML[&] | 0.94 | 22.1 | 15.5 | 110 | 2300 | 922 | 0.13/0.14 | 2.8 | 0.22/0.25 | 18 | 1.62 | 4.2 | 1.5 |
| PBL1 | 1.55±0.001 | 18.1±0.2 | 13.3±0.2 | 93±0.6 | 1600±70 | 717±0.42 | 0.16/0.16 | 3.2±0.3 | 0.24/0.25 | 28±2.5 | 1.58±0.02 | 4.1±0.3 | 2.2±0.4 |
| PBL2 | 2.9±0.2 | 10.5±1.5 | 9.4±0.5 | 76±3* | 2050±2300 | 248±37 | 0.14/0.14 | 1.3±1.6 | 0.19/0.20 | 8±2 | 1.60±0.02 | 2.2±0.4 | 0.6±0.1 |
| MT1 | 4.1±0.1 | 3.9±0.6 | 6.5±0.4 | N/A | 1532±68 | 112±20 | N/A/<0.1 | 0.31±0.1 | 0.20/0.25 | 3±2 | 1.57±0.04 | 0.8±.3 | 0.2+/0.1 |
| MT2 | 4.6±0.02 | 1.0±0.2 | 6.2±0.4 | 76±2 | 1515±720 | 125±36 | N/A /<0.1 | 0.4±0.4 | 0.20/0.20* | 3±2 | 1.65±0.02 | 0.5±0.2 | 0.15±0.1 |
| MT3 | 6.3±0.2 | -9±0.1 | 2.2±0.8 | 74±4 | 2893±1013 | 76+/12 | N/A /<0.1 | 0.2±0.5 | 0.10/0.12 | 1±1 | N/A | 0.2±0.1 | 0.1±0.1 |
| UT1# | 7.8±0.2 | -17.3±0.5 | 1.8±0.1 | 79±4 | N/A | N/A | N/A /<0.1 | N/A | N/A | N/A | N/A | 0.2±0.1 | 0.1±0.1 |
| UT2# | 8.5±0.1 | -21.6±0.2 | 0.9±0.2 | 80±2 | 8258±1192 | 62±10 | N/A /<0.1 | 0.1±0.1 | <0.1/0.12 | 1±1 | N/A | 0.2±0.1 | 0.1±0.1 |
| UT3# | 9.7±0.1 | -30.4±0.3 | 0.3±0.1 | 76±4 | 7687±1980 | 59±12 | N/A /<0.1 | 0.3±1.3 | <0.1/0.12 | 1±1 | N/A | 1±0.3 | 1±0.3 |
| UT4# | 10.5±0.2 | -38±2.3 | 0.4±0.1 | 78±1 | N/A | N/A | N/A /<0.1 | N/A | N/A | N/A | N/A | 0.6±0.4 | 0.2±0.1 |



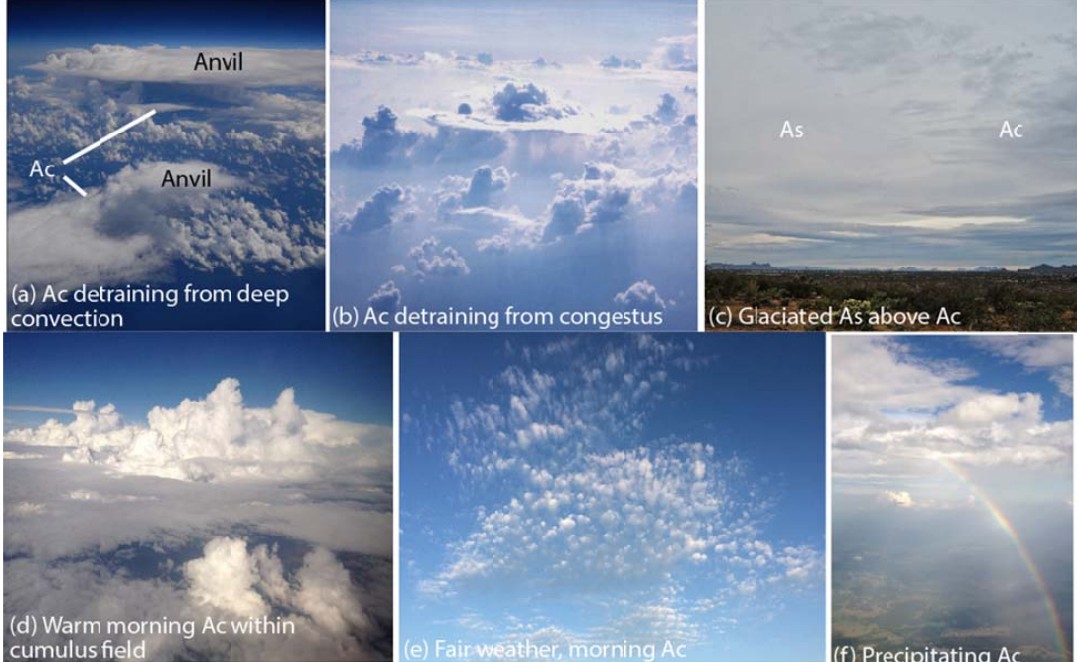

Figure 1. Cloud photographs of Ac and As characteristics. (a) Image from the NASA ER-2 showing Ac shelf clouds detraining from deep convection over the Gulf of Mexico during SEAC[4]RS;  (b) Ac detraining from cumulus congestus in a field of biomass burning smoke over Brazil (Reid et al., 1999); (c) mixed field of As above Ac clouds during a convectively active period in Arizona ; (d) Warm Ac clouds over developing cumulus field over west Texas during SEAC[4]RS ; (e) Morning fair weather Ac field over Monterey CA; (f) precipitating thin Ac clouds over central Texas during SEAC[4]RS (Photo credit, (a) S. Broce, NASA; (b) & (c)  A. Rangno, enhanced for contrast; (d) - (f), J. S. Reid).





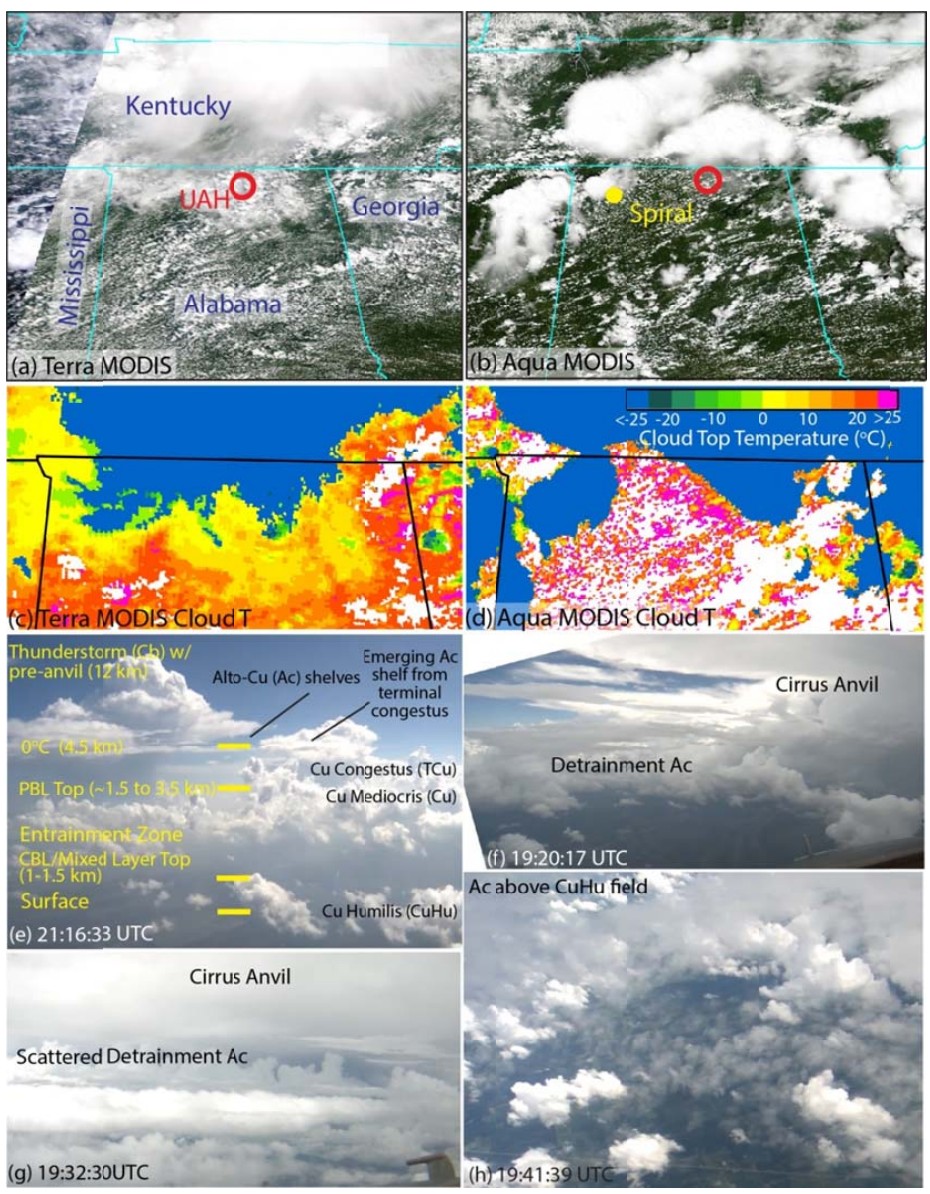

Figure 2. MODIS (a) Terra (16:00 UTC) and (b) Aqua (19:14 UTC) images, with markers indicating the location of the UAH lidar site (red) and DC-8 spiral (yellow) for August 12, 2013. Corresponding MYD06 cloud top temperatures zoomed onto northern Alabama are provided in (c) and (d). Also included are annotated camera images from the NASA DC-8 demonstrating cloud types (e) image just after profile components end; (f) forward images as the DC8 was about to enter a detrainment Ac at 4.4 km (g) forward image of the DC-8 while sampling 6.5 km aerosol layer; (h) nadir images of an Ac detrainment shelf exiting a Cb over a field of Cu.





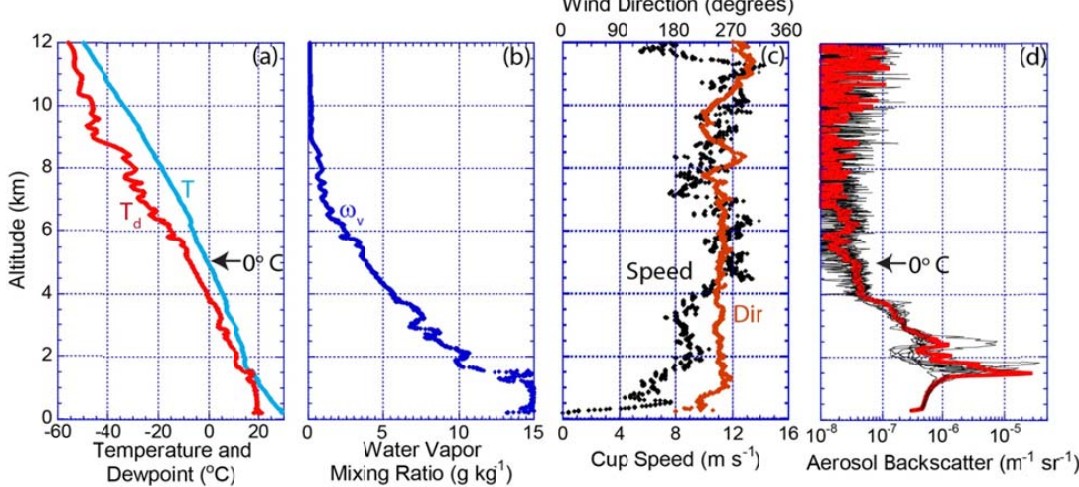

Figure 3. SEACIONS radiosonde release on August 12th, 2013 18:40 Z/13:40 CDT at Huntsville
(Altitude in MSL, 200 m greater than ground level). (a) Temperature and dewpoint; (b) water
vapor mixing ratio; (c) wind cup speed and direction. (d) 5 minute aerosol backscatter profiles
from the UW-HSRL at Huntsville for the two hours after the radiosonde release, with the mean
value in red.





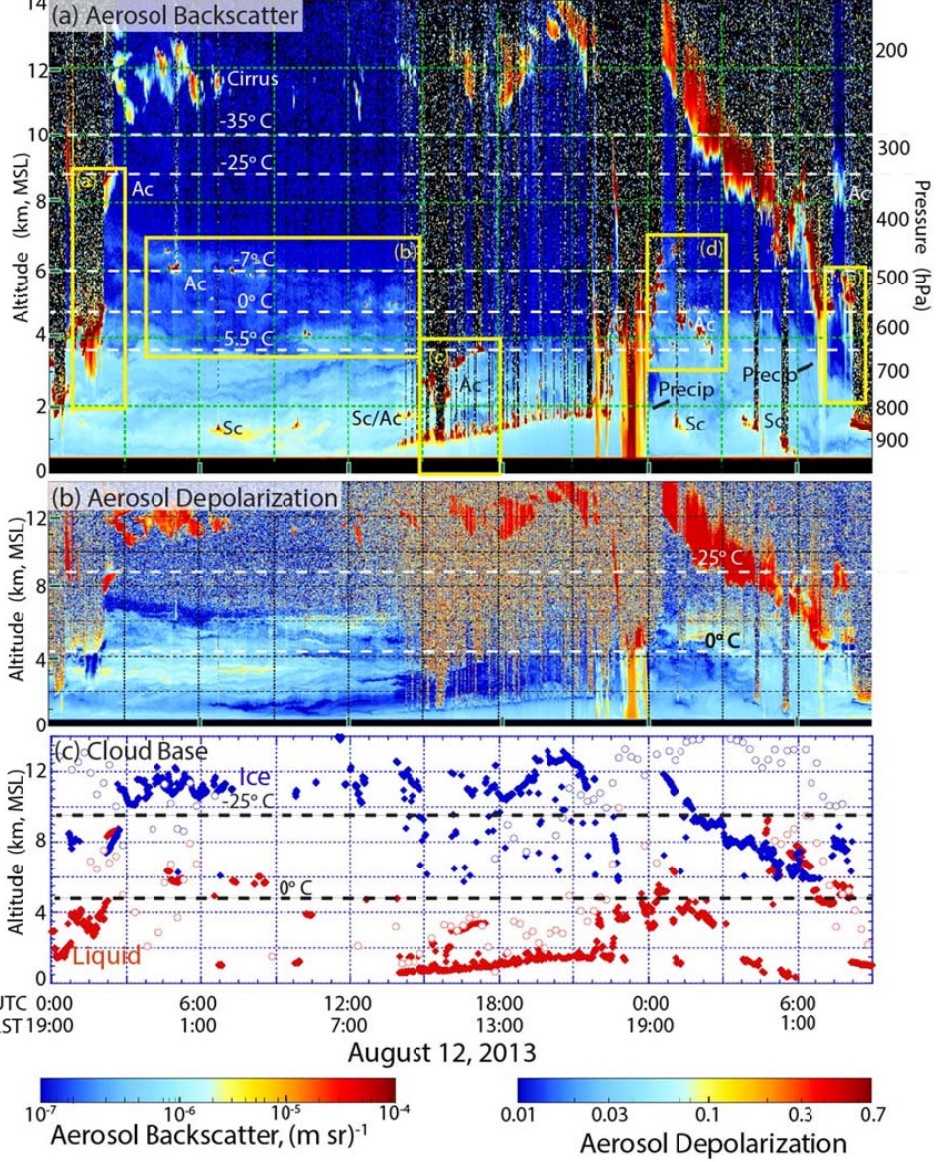

Figure 4. Example lidar data for August 12, 2013. UW HSRL aerosol (a) backscatter and (b)
depolarization from the surface to 14 km AGL. Listed are cloud types, phenomenon and, from a
13:30 radiosonde release, key temperature isopleths. Also shown in (c) are liquid and ice cloud
bases (solid) from the ground based HSRL, and liquid and cloud tops from GOES-13 (open). To
convert from AGL to MSL, subtract 220 m.

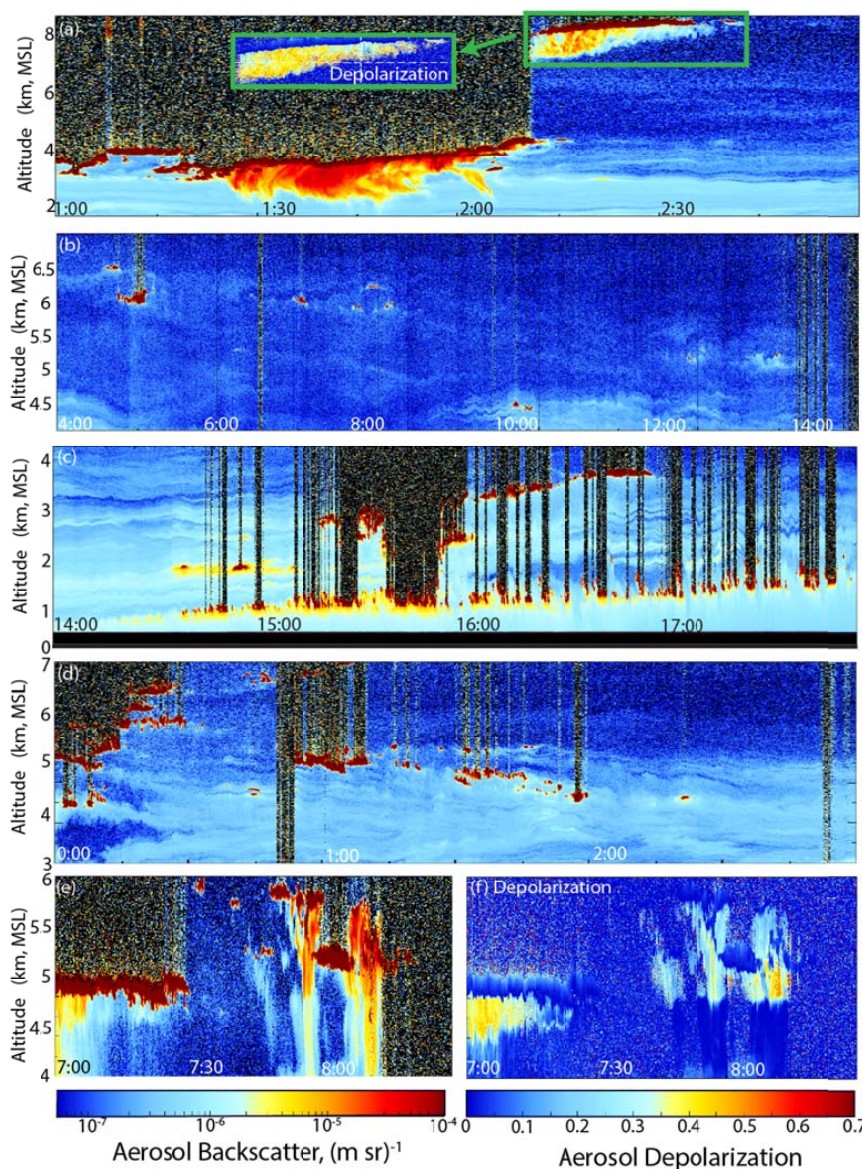

Figure 5. Aerosol backscatter for inset boxes as labeled in Figure 4 of key altocumulus and
aerosol features for the August 12, 2013 case. Included is aerosol depolarization where ice is
prevalent including an inset in (a), and a depolarization in (f) corresponding to (e). All times are
UTC.





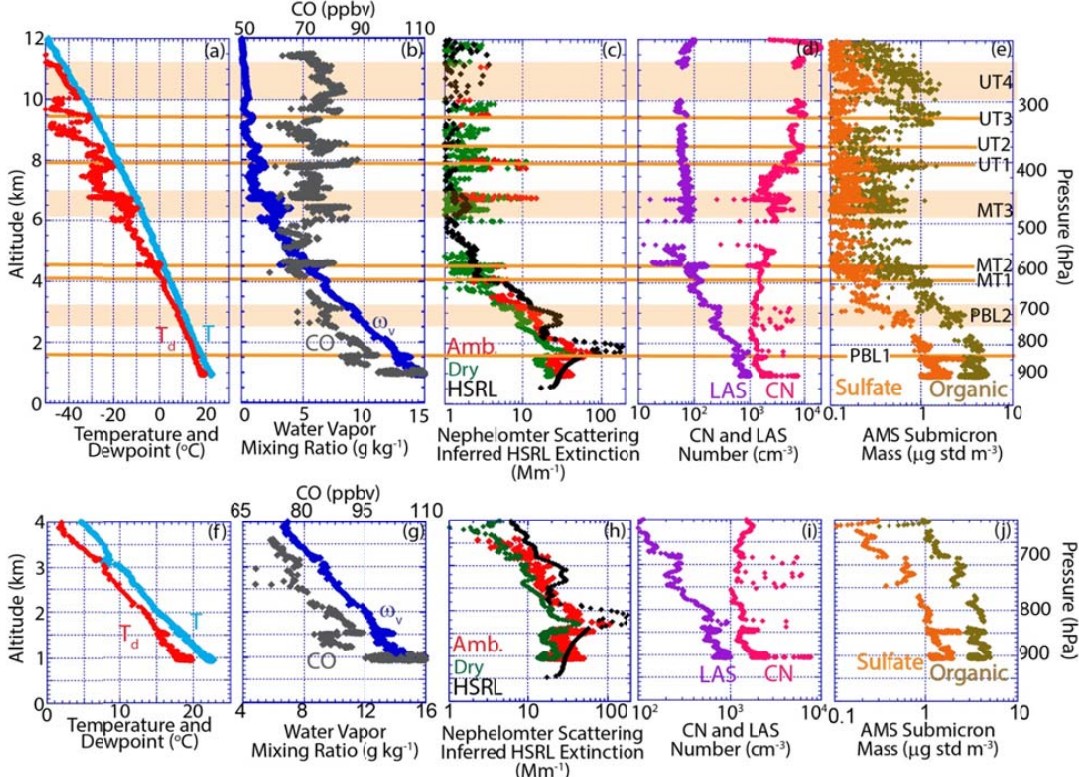


Figure 6. DC-8 aircraft spiral sounding data initiated at August 12, 2013 19:10:30 UTC on the
downwind side of a thunderstorm over northwest Alabama. Altitudes are relative to mean sea
level, ~ 300 m higher than AGL. Included is (a) Temperature and dew point; (b) Water vapor
mixing ratio ($\omega_v$) and CO; (c) DC-8 total ambient and fine dry 550 nm nephelometer with the
ground based UW HSRL derived extinction (lidar ratio=55 sr$^{-1}$) at Huntsville Al; (c) Number
concentration from laser aerosol spectrometer (LAS, $d_p$>0.1 μm) and condensation nuclei (CN,
dp>10 nm);  (e) Aerosol mass spectrometer organic materials and sulfate. Key moisture and
aerosol layers as discussed in the text are marked as orange lines or bands.(f)-(j), same as (a)-(e)
expanded in the vertical to enhance PBL feature readability and the legends are equivalent.





Figure 7. Time series of key meteorology, cloud and aerosol properties entering a detrainment
shelf cloud on Aug 12, 2014. The legends for the graphs on the left are the same for graphs on
the right




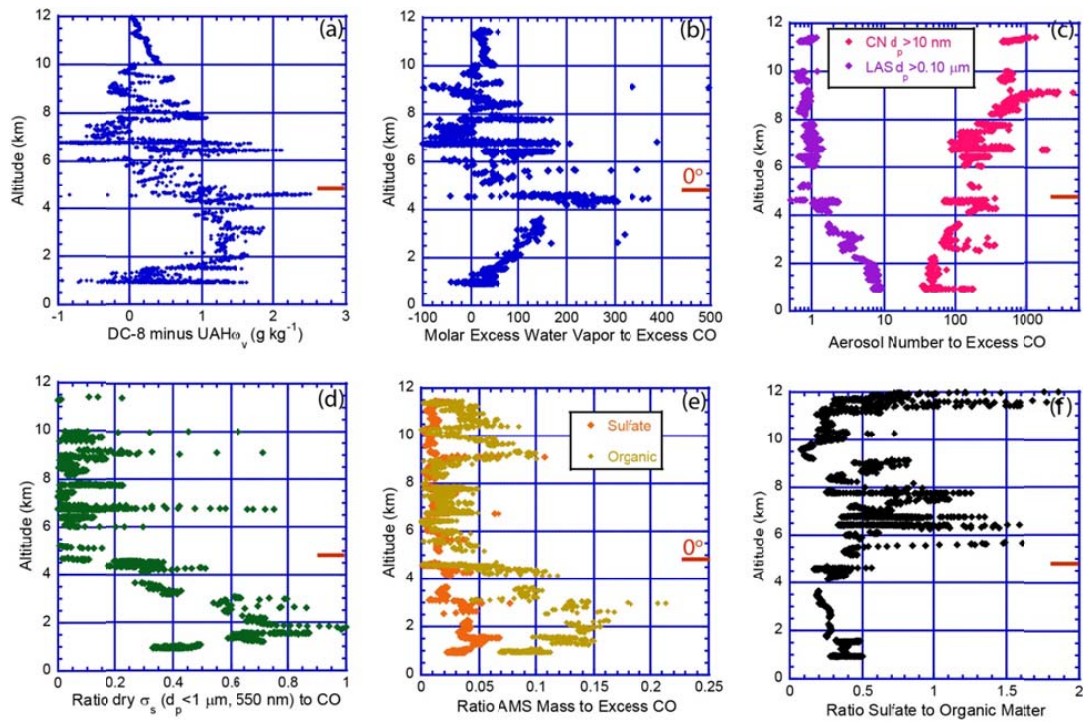



Figure 8. (a) Difference in water vapor mixing ratio between the DC-8 profile and the
SEACONS radiosonde release at Huntsville. Profiles of key constituents relative to excess CO
including (b) excess water vapor to excess CO; and (c) aerosol number; (d) dry light scattering;
(e) organic carbon and sulfate to excess CO; (f) ratio of sulfate to particulate organic matter.





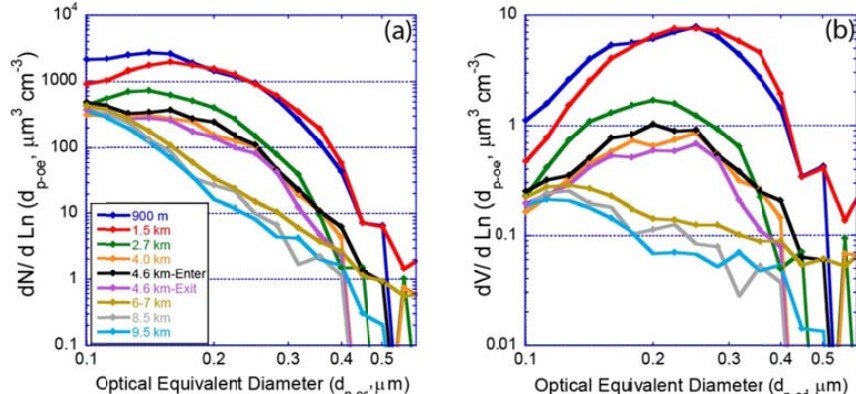


Figure 9. Laser Aerosol Spectrometer-LAS  (a) number and (b) volume distributions of aerosol
layers as a function of altitude. The legends for the two graphs are the same.





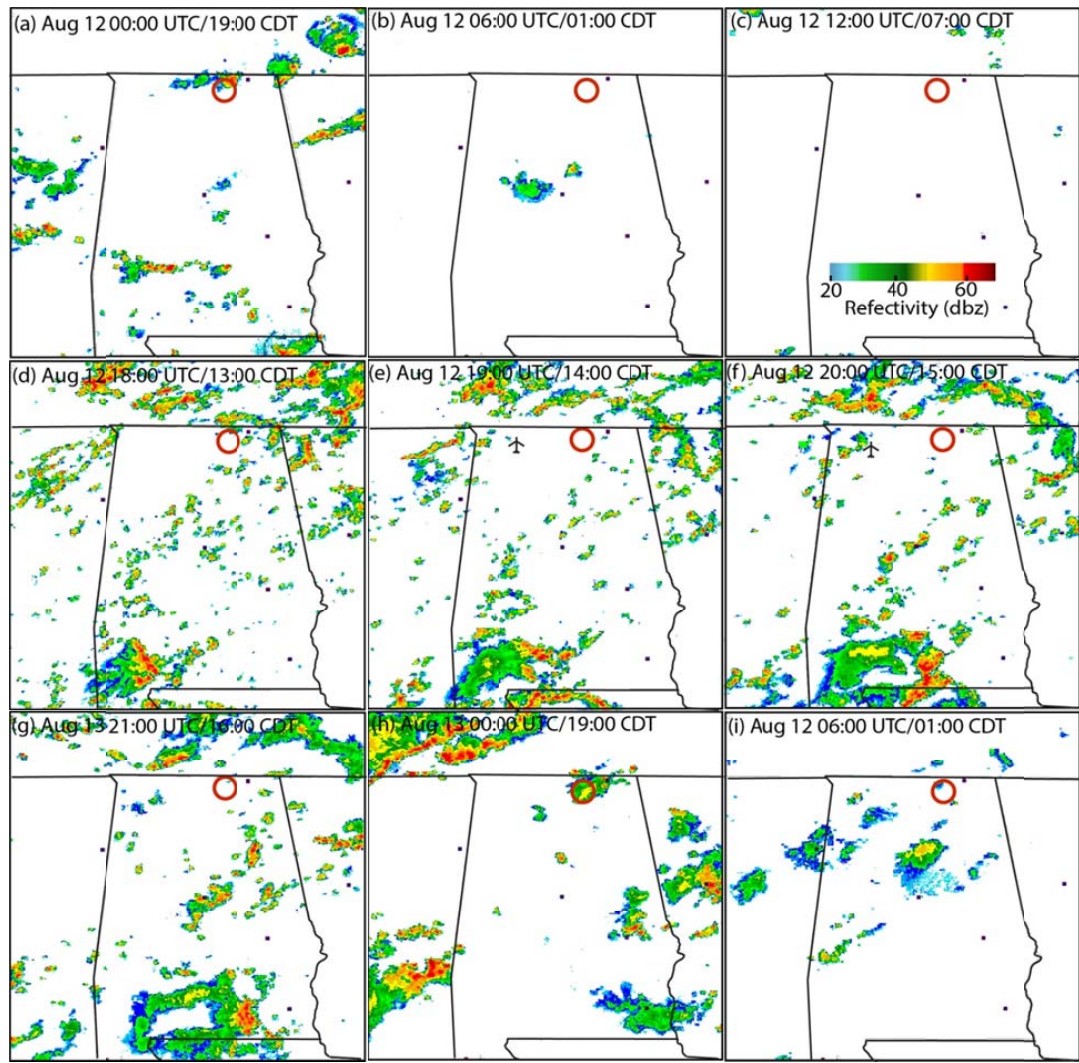


Figure A.1. NEXRAD radar reflectivity composites for August 12, 2013 study case. The red
circle indicates…… Pull the color bar outside so it is obvious that it can be used for all the
graphs???




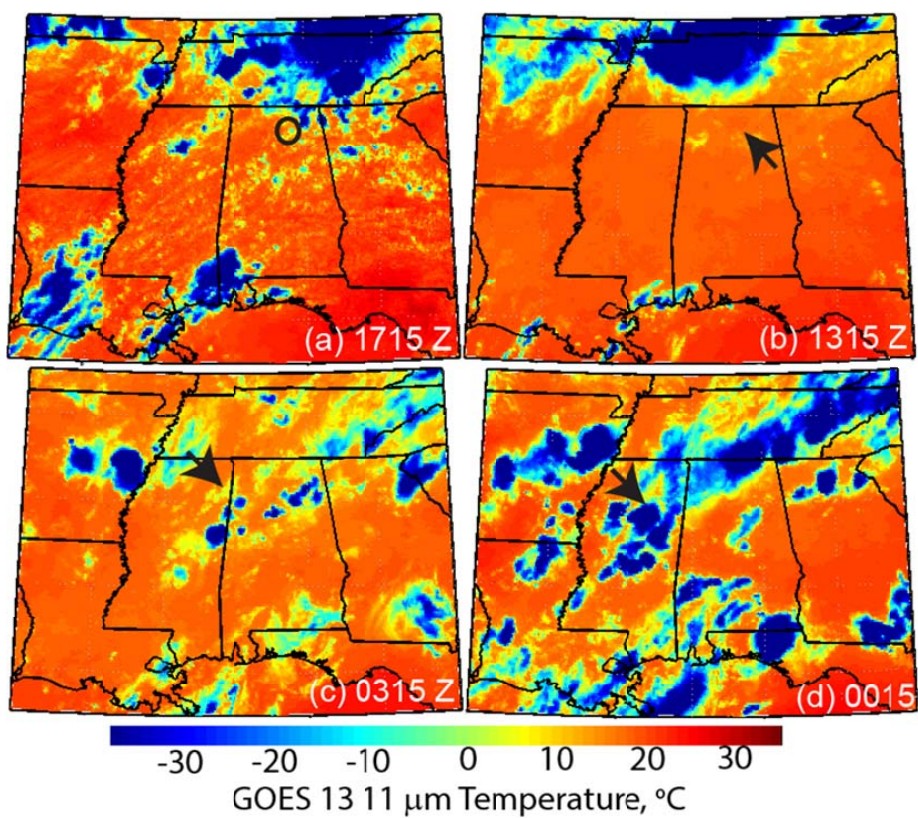


Figure A.2. GOES 13 11 μm channel images of the storm leading to AC clouds in the Huntsville
lidar data in Figure 5. (a) 12 Aug 2013, 1715z of PBL detrained AC clouds. (b) Initiation of back
trajectory for 0°C cloud. (c) 10 hrs back trajectory endpoint to large detrainment shelf (d) Cb that
formed the AC layer.









Figure A.3. Forward camera images from the DC-8 forward video taken from the leeward spiral
along the sampled thunderstorm on August 12, 2014 over northwestern Alabama




a) 19:34:22-24/70440

b) 19:34:24-27/70464

c) 19:34:34-19:38:36/70474

1.28 mm

d) 19:38:36-37/70716

e) 19:38:37-38/70717




106 Figure A.4. 2D-S images of ice for selected periods during layer sampling associated with the
107 right column of Fig. 7.. Temperatures were -~-10$^o$C, at an altitude of 6.75 km. Annulus are ice
108 imaged out of focus.