# Peer review of "Observations and hypotheses related to low to middle free tropospheric aerosol, water vapor and"

_Atmospheric Chemistry and Physics, 2019_

## Referee Comment (RC1) · Anonymous Referee #1 · 22 Apr 2019

**Review of "Observations and hypotheses related to low to middle free tropospheric aerosol, water vapor and altocumulus cloud layers within convective weather regimes: A SEAC4RS case study"**

This paper describes aerosol layers in the neighborhood of altocumulus clouds observed near Huntsville, Alabama on August 12th 2013, using a ground-based lidar and the NASA DC-8 aircraft. Based on a comprehensive analysis, the authors are able to make some tentative conclusions concerning the origin of the aerosols.

The analysis is comprehensive and the paper well-written. It is useful to analyse case studies in detail, especially in the difficult regimes studied here, and the paper is a valuable contribution.

My comments are all minor.

**Scientific comments**

I am reluctant to suggest adding further content to this already long study. However, it would be great to connect the aerosols to the clouds as tightly as possible, given the promises and inferences made in the introduction, abstract and discussion about the indirect effect. At the moment, the paper is mostly about cloud effects on aerosol but motivated to a large extent by aerosol effects on clouds. There is plenty of valuable qualitative discussion of the clouds present but not much quantitative information on cloud microphysics, which is surely critical to the climate effects of aerosols used to motivate the paper. It is repeatedly mentioned in the text that previous literature has found updraft speeds in these clouds to be low. Presumably, these updraft speeds were actually measured by the DC8 aircraft on the days in question, so can they be presented alongside the existing aerosol concentrations, and ideally CDP cloud droplet number, to speculate on aerosol activation? I appreciate that detailed cloud modeling is not in the scope of the study, but perhaps typical updrafts here could be compared with those found in closure studies in other cloud types (e.g. Guibert et al, JGR 108 8628 (2003); Snider et al, JGR 108 8629 (2003)) to speculate more precisely about the supersaturations and activation diameters of aerosol these clouds might require? Then Hypothesis 6.1 could really be linked more explicitly to the previous sections.

L350 would be helpful to specify the altitude of the melting level
L421 would be helpful to specify approximately how thick the layer is (or if 1km, then how high it is).
L431-4 Could the entire flight track be marked on one of the plots?
L507 The increase in humidity seems quite small when one compares the temperature and dewpoint, but it is probably just my failure to eyeball the plots correctly. Would be helpful to put some numbers in here to quantify how much the humidity increases.
L514 Is "consecutive" the right word here? Simulataneous? And it looks like the nephelometer spike is a touch above the spike from the counters, is that due to sampling line delays or a real effect?
L517 Please specify what is believed to be detraining. SO2, organics, or particles, perhaps?
L594 "they are not directly observed at Huntsville" For MT3 this seems inconsistent with Figure 5c and line 623 of the text.

L733 The phrasing could be improved here – I think the secondary mass should help particles activate and then be nucleation scavenged, but the secondary mass shouldn't be stripped from the particles by the cloud, which is somehow implied here.

L750. I agree with sulfate being produced by homogeneous nucleation, but SO2 tends to be found at relatively similar concentrations at all altitudes in the troposphere, while organics and nitrates and so on decrease in concentration with altitude. I think this is most likely responsible for the increased sulfate fraction in the upper troposphere.

L773-6 Perhaps worth mentioning some more recent work here, for example on tenuous warm low clouds, e.g. Wood *et al* 2018, https://journals.ametsoc.org/doi/abs/10.1175/JAS-D-17-0213.1 Similarly at line 801, are there parallels with the nucleation seen in pockets of open cells, eg. Kazil *et al* (2011), https://www.atmos-chem-phys.net/11/7491/2011/

**Textual suggestions**

There are a few places where it might be possible to aid readability by better signposting the content of each paragraph in the first sentence. The paper is long and sometimes it's easy to lose sight of the big picture while reading the details..

L107 suggest Ac "formation" by mesoscale lifting
L134 suggest "to examine"
L297 suggest remove comma
L321 "n"->"in"
L364 "phenomenon"->"phenomena"
L369 "was"->"were" or "their"->"its"
L382 "days"->"day's"
L390 can the individual pockets of aerosol be pointed out on the plots?
L452 and 455 Figure 5->Figure 6
L457 "was"->"were"
L505 "combination"->"combination of"
L569 "detriment"->"detrainment"
L628 "actually"->"actual"
L632 "observations"->"observation"

The paragraph starting at line 651 could be swapped with the next one, so the reader learns or is reminded why the water vapor to CO ratio is of such paramount importance as soon as it is mentioned, instead of waiting a paragraph.

L698 "contently"->"consistently"?
L718 "precursors"->"precursor"
L722 "provides"->provide
L728 "exiting" the clouds? Or "existing"
L734 "evidence"->"evidenced"
L782 "high"->"highly"
L875 remove second instance of "consistent with"
L883 "model"->"mode"
L902 "Parungno"->"Parungo"
L983 "imbedded"->"embedded"
Figure A1 caption contains a comment that is unlikely to be intended for publication.

---

## Referee Comment (RC2) · Anonymous Referee #2 · 4 Jun 2019

The paper addresses covariability of cloud, water, and aerosol features for mid-level Altocumulus (Ac) clouds in a detailed case study based on observations during the August 2013 NASA SEAC4RS airborne campaign. Observations from high sensitivity ground-based lidar are combined with measurements from the NASA DC-8 aircraft look at residual from a storm system near Huntsville, AL, on August 12, 2013. The complex variability of aerosol and trace gas species along with water vapor, cloud features, and thermodynamic profile suggest several hypotheses regarding relationships of aerosols

and Ac clouds.

The paper is well written and thorough. Aside from a few textual suggestions below I have only a minor comment that the paper could include some further statement of the significance of studying this phenomena, and would welcome some further speculation on approaches. The introduction points to a paucity of literature on Ac, but also cites literature that speculates on relationships between Ac clouds and increasing aerosol burden in the future. It seems significant that 30% of cloud area fraction is characterized as Ac. But we have no estimate on the aerosol-cloud radiative effect for this class of phenomena? Is that a significant hole in our understanding of aerosol indirect effects? What kind of observing system would it recommend? The authors are well aware of the ongoing NASA Aerosol & Clouds, Convection (ACCP), and Precipitation study being carried out in response to the NRC Decadal Survey. Where does observing these systems fit into overall goals for ACCP? What observational demands does it imply? Is there a particular form of space-based lidar that contributes to this study, or it hopeless to do from space? Or do systematic observations from selected ground sites with aircraft and ground-based lidar, along with high resolution modeling, hold the key to unraveling these systems and addressing the hypotheses posed?

Minor points:

76: Two references to Warren et al. 1986. Maybe Warren et al. 1988 for one of them?

182-185: Could you put these numbers in context of CALIOP, just to give a sense of how hard this will be to do from space?

321: "in the next section"

344: I don't think omega_v has been defined at this point, but I see it is later (450).

473: "trace," not "tracer"

698: "content," not "contently"

Figure A.1: read and correct the caption.

Figure A.3: "images," not "mages"
* * *

---

## Author Comment (AC1) · 16 Jul 2019

We would like to start by thanking reviewer 1 for taking the time to review the manuscript. This paper was a result of a realization that to mine the entire SEAC4RS dataset on aerosol-Ac related issues, we needed a frame of reference. There simply was not a good paper in the literature that demonstrates the many facets of the phenomena. The August 12, 2013 flight was the one flight where we could see everything

going on. In response to your major overarching comment that it would be beneficial to the paper if we could focus more on the aerosol impact on clouds, we heartily agree, but simply do not have space to do it here. This is especially true that we only have one really good Ac pass and these clouds are only a hundred meters or so deep. So it is very difficult to perform the CCN analysis as suggested on a single case. Remote sensing of Ac is also difficult because of their fine cellular nature. Co-author Posselt and I have been devising strategies on how to best model these Ac clouds as it is quite tricky. The aerosol field is imbedded in the detrained cloud layer. There is some hope in that the cloud formation is at the very top of the layer. But it is for these reasons that this phenomenon paper we focused on the covariablity and vertical structure. The next paper we are currently constructing has over a half dozen other cases, but the aerosol-cloud microphysics relationships are anything but clean-cut. We have expressed your points in the current draft of the paper in Section 6 and a new Section 7. As mentioned by reviewer two, the aerosol-Ac problem is something that is worthy of a great deal of attention by the community. It is a focus area of the upcoming CAMP2Ex mission, and has garnered the attention of many in the ACCP community.

Specific comments. L350: On melting level heights, 4.5-5 km. Added

L421: "would be helpful to specify approximately how thick the layer is" I am afraid I don't understand. The sentence specified 200-300 m in depth. Which thickness are you looking for?

L431-4: On adding flight tracks. Adding flight tracks to Figure 2 looks messy, but we added the full flight track as a new figure, Figure A.1.

L507: On RH fields, Added "80% mid mixed layer reaching ∼90% between clouds." We did not really focus on the PBL in this paper, as the way the flight path was conducted with the strong gradient in mixed layer properties in the lead up to the profile suggested samples were aliased down there.

L514: Thank you simultaneous is better. . ..

L517: You are being more specific, "This enchantment is presumably through the detrainment of mixed layer air via the fair weather cumulus." We pointed out "Also (not shown) was a likewise spike in SO2 and NO2 to roughly mixed layer levels (10's-> 100's of ppbv), and a minor dip in ozone. (40->37 ppbv)." But we do not wish to go into details on the gas chemistry here.

L594 "'they are not directly observed at Huntsville' For MT3 this seems inconsistent with Figure 5c and line 623 of the text." Sorry for the confusion. Our point is that these are not the exact same layers whereas the mixed layer is by definition the same layer at Huntsville. We changed the subsequent text to "These layers are similar in nature to layers observer throughout the day at Huntsville."

L733 "The phrasing could be improved here – I think the secondary mass should help particles activate and then be nucleation scavenged, but the secondary mass shouldn't be stripped from the particles by the cloud, which is somehow implied here." Yes, we did not mean to imply that at all, rather the cloud/precipitation process is a net reduction from the secondary mass production. Corrected to "However these same aerosol particles that grow to larger sizes are more likely to be lost to droplet nucleation and scavenging. "

L750. "I agree with sulfate being produced by homogeneous nucleation, but SO2 tends to be found at relatively similar concentrations at all altitudes in the troposphere, while organics and nitrates and so on decrease in concentration with altitude. I think this is most likely responsible for the increased sulfate fraction in the upper troposphere." We agree with the reviewer that in general the overall nucleation mode may well be a regional background, but we have also found in our own measurements and the literature that we can find increasing and decreasing SO2 and sulfate with height. Prompted by this comment, we look back and find that CN in particular is anti-correlated with water vapor, suggesting that CN may be due to background. But higher sulfate mass in the mass spectrometer is sometimes positively correlated with water vapor and OC mass. A good compare and contrast is UT1 and 4 (high sulfate), versus UT 2 and 3(Low sulfate). We think that overall throughout the column significant amounts of homogeneous nucleation, but the mass is still a result of some form of cloud processing. Indeed, we find a drop in $SO_2$ at the locations of detrainment. We mention this now in the paper, although we would prefer to leave a more detailed paper on gas chemistry outside of CO for a separate paper. But we now mention this in the paper, and added $SO_2$ and $CO_2$ to our our figures.

L773-6 "Perhaps worth mentioning some more recent work here, for example on tenuous warm low clouds, e.g. Wood et al 2018, https://journals.ametsoc.org/doi/abs/10.1175/JAS-D-17-0213.1 Similarly at line 801, are there parallels with the nucleation seen in pockets of open cells, eg. Kazil et al (2011), https://www.atmos-chem-phys.net/11/7491/2011/" Good suggestion!

Textual suggestions: All corrected. Thank you very much for the proof read.

---

## Author Comment (AC2) · 16 Jul 2019

We very much appreciate reviewer 2 taking the time to review the paper late in the game and provide valuable feedback. We agree that that Ac decks in general, and the aerosol-Ac system in particular, need quite a bit more attention in the community. As we remarked to reviewer 1, this system will be investigated in CAMP2Ex, and has received some attention on the ACCP team (co-author Trepte is a SLAT participant).

[Figure]

With your encouragement, we have added an additional paragraph to the discussion to more clearly address some of the observation issues. Perhaps part of the reason why the Ac system is so neglected is the difficulty in making even a simple observation? I think Parungo lays out the logic quite well in her 1996 paper. But between the thin cellular nature and the proclivity to form along even minor inversions leads to serious remote sensing and sampling challenges, let alone developing a modeling frame work which can account for these creatures. We explore this much more fully in the added paragraph.

Text edits: Corrected as suggested, thank you.

182-185: "Could you put these numbers in context of CALIOP, just to give a sense of how hard this will be to do from space?" We added the following text: "The UW-HSRL was able to extract the aerosol backscatter profile to very high fidelity. Unlike more common elastic backscatter lidar measurements that must de-convolve a combined molecular and aerosol signal in an inversion, HSRL systems can separate a line broadened molecular backscatter signal from the total backscatter signal via a notch filter (Eloranta et al., 2005, 2014; Hair et al;, 2008). The difference is used to calculate aerosol backscatter. For this deployment the UW HSRL performed with a precision in aerosol backscatter of better than 10-7 (m sr)-1 for a 1 minute average, and 10-8 (m sr)-1 for 15 minute averages. In comparison, Rayleigh backscattering is 1x10-6 (m sr)-1 at 4 km, and 5x10-7 (m sr)-1 at 10 km. Thus at 15 min averaging, precision is likewise better than 1 to 5% of Rayleigh. This very high sensitivity to aerosol scattering is a result of the combination of the aforementioned HSRL ability to separate the molecular from aerosol scattering, the large signal to noise of the instrument, and the high solar background rejection during daytime observations. It is challenging to make a direct comparison of the ground based HSRL to CALIOP given the very different viewing geometery and sampling combined with the highly variable SNR of CALIOP between day/night observations. The NASA Langley airborne HSRL was used to validate the CALIPSO aerosol retrievals (S.P Burton et al. 2013) and found that only 13% of the

layers identified as smoke by the Langley HSRL was correctly identified by CALIOP using the V3 CALIOP products. The UW HSRL, being a stationary ground-based system, provides even greater sensitivity to the aerosol backscatter as it can dwell over the same location for a long period of time.